# Causal Regression: Learning Causal Mechanisms for Robust and Interpretable Prediction

## Abstract

The performance of standard regression models, which primarily learn statistical associations, is vulnerable to label noise. This paper proposes Causal Regression, a paradigm that shifts the focus toward learning invariant causal mechanisms. We introduce CausalEngine, a neural architecture that operationalizes this paradigm based on the Distribution-consistency Structural Causal Model (DiscoSCM). It first performs abduction to infer a distribution over latent cause, and subsequently applies a causal mechanism to make a prediction. The mathematical properties of the Cauchy distribution facilitate an analytical inference process. This design sidesteps the need for sampling-based approximations, thereby eliminating the high-variance gradients and computational overhead they introduce, leading to stable and efficient end-to-end training. This design also provides a structured form of interpretability by decomposing predictive uncertainty into two distinct sources: epistemic uncertainty, arising from incomplete knowledge of an individual, and aleatoric uncertainty, stemming from inherent environmental randomness. Our experiments demonstrate CausalEngine's significant robustness against label noise. Especially in high-noise regimes where strong baselines falter, our approach exhibits a significantly smaller drop in performance. This work suggests that shifting the modeling focus from statistical associations to causal structures is a promising direction for building AI systems that are more reliable and interpretable.[1]

## 1 Introduction

When one mentions regression analysis (Hastie et al., 2009; Bishop, 2006), the California housing dataset (Pace & Barry, 1997) often comes to mind: a collection of features about districts, and a house price to predict. When one mentions causality, it almost invariably involves a "treatment" from settings such as a clinical trial or a marketing campaign (Pearl, 2009; Neyman, 1923; Rubin, 1974; Holland, 1986). Its core lies in an intervention, a concept seemingly absent in purely observational data like the housing dataset. This paper introduces **Causal Regression**, a novel regression algorithm framework rooted in causality. An immediate and unavoidable question arises: for datasets of this kind that lack any explicit "treatment", what does it actually mean?

To answer this, we need to view "features" from a new perspective. Specifically, while traditional regression treats all features as a flat list of predictors, Causal Regression posits that observed variables are generated by a set of more fundamental causal attributes. For instance, for the housing dataset, this (latent) causal variable $U$ might represent a district's intrinsic *community quality*, *development potential*, or *school district prestige*—the true drivers of housing prices. The features we observe, such as median income or the local crime rate, are merely projections generated by this deeper causal variable. Therefore, our Causal Regression on this dataset is not about finding a treatment; it involves **inferring the (unobservable) causal variable** $U$. While this task falls under the broad umbrella of causal discovery (Spirtes et al., 2000), our objective is fundamentally different: instead of discovering the complete causal graph, we aim to learn a robust predictive model by uncovering the latent causes of the target variable.

Our Causal Regression transforms prediction from a paradigm in associational learning into one of causal modeling. Instead of estimating a conditional expectation from correlations, we first infer

---

[1]Code is available at `https://anonymous.4open.science/r/causal-regression-135C`.

the (unobserved) cause $U$ and then use a structural equation to model how this cause generates the outcome. Inferring a (latent) cause —-a process technically known as **abduction**—- is a core step of counterfactual reasoning theory. To rigorously capture individual heterogeneity, we ground this process in the DiscoSCM (Gong et al., 2024), which explicitly decouples stable individual factors from random noise. Specifically, we introduce the **CausalEngine**, a four-stage architecture that implements this causal learning framework. It consists of four stages, see Figure 1:

1. **Perception**: Extracts a representation $Z$ from the input features $X$.
2. **Abduction**: Infers the underlying cause $U$ from the representation $Z$.
3. **Action**: Computes a decision score $S$ from the cause $U$ via causal mechanisms.
4. **Decision**: Makes a final, task-specific prediction $Y$ based on the causal score $S$.

**CausalEngine's Four-Stage Process**

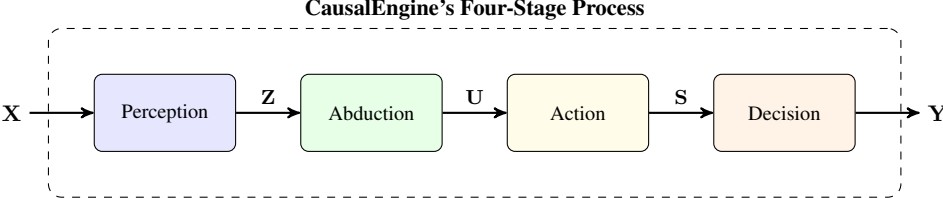

Figure 1: The architecture of the CausalEngine. It transforms input features $\mathbf{X}$ into predictions $\mathbf{Y}$ through a four-stage causal reasoning process: Perception, Abduction, Action, and Decision.

The explicitly structured architecture of CausalEngine endows it with interpretability-by-design, a significant departure from prevalent post-hoc explanation techniques (e.g., LIME, SHAP) (Ribeiro et al., 2016; Lundberg & Lee, 2017). While these methods provide external, often approximate, attributions for a model's predictions, CausalEngine's process is inherently transparent. It is analogous to a physician's diagnostic reasoning: observing symptoms (Perception), assessing the range of potential causes (Abduction), and then determining a course of treatment (Action). This transparency, afforded by direct access to internal states like the posterior of $U$, is a core feature. While conceptually elegant, this transparent architecture presents a practical challenge: how to ensure both computational tractability and end-to-end learnability. Our solution hinges on two foundational choices: modeling the latent cause with the Cauchy distribution and enforcing a linear structure on the causal mechanism, which enables analytically tractable inference without sampling. This stands in contrast to the prevailing methodology in Deep Structural Causal Models (DSCMs) (Pawlowski et al., 2020; Poinsot et al., 2024), which commonly employs non-linear functions for the causal mechanisms, often underpinned by Gaussian distributional assumptions.

The rationale for this contrasting approach is rooted in the foundational concept of the latent variable $U$, whose dual role as both an *individual selection variable* and an *individual causal representation* is formally established in (Gong et al., 2024). From this foundation, our methodology is built upon two core hypotheses:

- **The Assumption of Linear Causality.** Inspired by the success of reward modeling in large language models (Ouyang et al., 2022; Zhong et al., 2025)—where a simple linear head over a powerful representation can capture the immense complexity of human preference—we hypothesize that causal mechanisms are fundamentally simple. The challenge lies not in learning complex structural equations, but in learning a sufficiently powerful representation that linearizes the problem.

- **The Language of Causality as Cauchy Distribution.** In line with the Potential Outcome framework (Imbens & Rubin, 2015) -—where outcome randomness is attributed to the selection of heterogeneous individuals—we contend that the Gaussian distribution is misaligned, as its light tails effectively foreclose extreme possibilities. We therefore adopt the Cauchy distribution, whose heavy tails mathematically embody the epistemic humility required to ensure that any outcome remains possible for any individual, a cornerstone of "everything is possible" in a counterfactual world.

While the predictive power of neural networks stems from their ability to fit complex statistical correlations (Rumelhart et al., 1986; Goodfellow et al., 2016), this very reliance on observational

patterns is the direct cause of their two most critical limitations: brittleness and opacity (Szegedy et al., 2013; Adadi & Berrada, 2018). Instead of fitting these spurious correlations, CausalEngine models the underlying invariant causal mechanisms. The superior robustness of our approach is demonstrated through extensive experiments. Across various benchmarks, CausalEngine shows remarkable resilience to data corruption, maintaining stable predictions where conventional models fail. For instance, even when 40% of the training labels are corrupted by systematic noise, our model's performance degradation is minimal, achieving a relative improvement in prediction accuracy of over 25% compared to the best-performing baseline. Furthermore, its inherent causal structure provides transparent interpretability by design, moving beyond the limitations of post-hoc explanations. This causal regression framework therefore provides a novel approach towards building predictive models that are more reliable and trustworthy.

To this end, this paper makes the following major contributions:

1. **A New Robust Regression Framework.** We propose a new framework that shifts regression from modeling brittle statistical associations to invariant causal mechanisms. This focus on causality yields models with significant robustness and extends causal inference to conventional regression problems.

2. **An Interpretable-by-Design CausalEngine.** We introduce CausalEngine, a novel four-stage architecture that operationalizes the causal regression framework. Its explicit reasoning process provides inherent interpretability, moving beyond post-hoc explanations.

3. **Assumptions for Causal Learning.** We introduce two key assumptions: the linear causal mechanisms and the use of Cauchy distribution for abduction. This theoretical framework enables an analytically tractable propagation of uncertainty from abduction to prediction, obviating the need for heuristic sampling-based methods.

The remainder of this paper is organized as follows. Section 2 lays out the theoretical foundations of Causal Regression, motivating the shift from traditional paradigms and providing a formal definition of this new approach. Subsequently, Section 3 presents the CausalEngine architecture in detail as a concrete implementation of the Causal Regression framework. In Section 4, we conduct a comprehensive empirical evaluation, testing our model's robustness and performance against established baselines on both synthetic and real-world data. Finally, Section 5 concludes the paper and discusses future research directions. Before delving into the details of our work, we first discuss the related literature.

## 1.1 RELATED WORK

**Learning Structural Causal Models.** Deep Structural Causal Models (DSCMs) (Pawlowski et al., 2020; Xia et al., 2021; Poinsot et al., 2024) leverage deep generative models to learn causal mechanisms, yet face immense challenges (Poinsot et al., 2024; Sick & Dürr, 2025). The primary difficulty is the strong dependence on a known causal graph $G^\star$ (Poinsot et al., 2024): without this structural constraint, causal mechanisms are fundamentally unidentifiable (Pearl, 1995; 2009). Even with a known graph, hidden confounders introduce spurious correlations that bias counterfactual estimations; while some methods can handle them (Xia et al., 2021; Rahman & Kocaoglu, 2023; Parafita & Vitrià, 2020; Sauer & Geiger, 2021), they still require knowing the exact position of unobserved variables.

Can causal discovery algorithms circumvent this limitation? Unfortunately, these methods rely on strong, often untestable assumptions (e.g., causal sufficiency, faithfulness) and suffer from high sample complexity and instability (Spirtes et al., 2000; Glymour et al., 2019). We argue for a different approach grounded in DiscoSCM (Gong et al., 2024). As shown in Table 1, DiscoSCM explicitly decouples the individual selection variable $U$ from exogenous noise $E$, allowing abduction to infer a stable individual type rather than random noise. By directly inferring this individual causal representation $U$, we circumvent the difficult problems of causal identification and graph misspecification.

**Robust Regression Methods.** A rich literature on robust regression addresses the challenge of stable prediction in the presence of outliers. One family redesigns the loss function: M-estimators like the Huber loss interpolate between quadratic and absolute losses to mitigate outlier influence (Huber, 1964), while Quantile Regression uses an asymmetric pinball loss to model conditional quantiles (Koenker & Bassett, 1978). A second family derives robustness from algorithmic structure:

Table 1: Comparison of Causal Frameworks. DiscoSCM's decoupling of individual $U$ and noise $E$ enables CausalEngine to learn stable individual representations.

| Feature | Potential Outcomes (PO) | Structural Causal Models (SCM) | DiscoSCM (Adopted) |
|---|---|---|---|
| **Core Primitive** | Potential Outcome $Y(t)$ | Structural Equation $v_i \leftarrow f_i(pa_i, u_i)$ | Individualized Eq. $v_i \leftarrow f_i(pa_i, e_i; u)$ |
| **Individuality** | Explicit in unit index | Confounded with exogenous noise $u$ | Decoupled: $U$ (individual) vs. $E$ (noise) |
| **Abduction Target** | Sub-population | Unobserved Noise $u$ | posterior of individual selection variable $U$ |
| **Key Advantage** | Explicit Individual Semantics | Unified Causal Ladder | Tractable L3 Inference via $U$ |

tree-based ensembles such as Random Forests (Breiman, 2001) and Gradient Boosting (Friedman, 2001) naturally isolate outliers through recursive partitioning.

However, these methods are designed for **statistical robustness**—producing reliable predictions by being insensitive to data perturbations—rather than modeling the underlying data-generating process. Consequently, they do not provide insights into causal mechanisms, a limitation our work addresses.

## 2 THE CAUSAL REGRESSION FRAMEWORK

The conventional starting point for prediction tasks is statistical regression, which aims to model the conditional expectation $E[Y|X]$. This approach, which has achieved tremendous success on independent and identically distributed data, is fundamentally designed to capture *correlation*, not *causation* (Pearl, 2009). This reliance on statistical association is brittle. When faced with distribution shifts, confounding factors, or adversarial perturbations, a model that has only learned spurious correlations can see its performance degrade sharply (Arjovsky et al., 2019). Moving beyond such surface-level associations to learn the invariant *causal data-generating process* offers a promising path toward genuine robustness (Peters et al., 2016; 2017).

To rigorously derive our framework, we rely on the **Population-Level Valuation** theorem established in DiscoSCM (Gong et al., 2024). Extending the intuition of Pearl's Abduction-Action-Prediction (Pearl, 2009), this theorem proves that any identifiable causal query can be computed through a structured three-step mathematical derivation:

1. **Abduction**: Infer the posterior distribution of the latent individual variable $P(U|X)$ from observed features.

2. **Valuation**: Evaluate the structural equations given the inferred individual $U$.

3. **Reduction**: Marginalize over the exogenous noise to obtain the final expectation.

Inspired by this "Abduce then Predict" procedure, we propose a new framework: **Causal Regression**, which represents a fundamental shift from learning a direct mapping $X \to Y$ to modeling the underlying causal mechanisms $U \to Y$. The cornerstone of this framework is the structural equation:

$$Y = f(U, \varepsilon) \tag{1}$$

where the components are defined as:

1. $U$ **(Latent Causal Representation):** A representation of the latent causal factors that are the true drivers of the outcome.

2. $f$ **(Causal Mechanisms):** A set of structural equations that maps the causal representation to the outcome. It is assumed to be stable and invariant across various environments.

3. $\varepsilon$ **(Exogenous Noise):** A random variable representing environmental randomness or measurement error.

In other words, this framework reframes the prediction task around a central hypothesis: the observed features $X$ are not the direct causes of the outcome $Y$. Instead, both are manifestations of a more fundamental set of unobserved, latent causal variables $U$. To operationalize this, our framework decomposes the problem into two distinct, learnable stages:

1. **Abduction:** The first stage reasons backwards from observed evidence ($X$) to infer a distribution over plausible underlying causes ($U$). This is performed by an **Abduction Model** ($g$), formally defined as $g : \mathcal{X} \to \mathcal{P}(\mathcal{U})$.

2. **Prediction:** The second stage uses the inferred cause ($U$) to predict the outcome ($Y$). This is performed by a **Causal Mechanism ($f$)**, which embodies the invariant predictive logic and is formally defined as $f : \mathcal{U} \times \mathcal{E} \rightarrow \mathcal{Y}$.

These two models are learned jointly by optimizing a unified objective function. The objective is designed to ensure that for a given input $X$, the predicted outcome, derived by first inferring $U \sim g(X)$ and then applying $Y = f(U, \varepsilon)$, accurately reconstructs the observed outcome. The specific architectural choices for $g$ and $f$, along with the formulation of the learning objective that makes this framework tractable, constitute our proposed **CausalEngine**, which is detailed in the subsequent section.

## 3 THE CAUSALENGINE

The Causal Regression paradigm, centered on the structural equation $Y = f(U, \varepsilon)$, presents a fundamental shift from correlational to causal modeling. However, translating this principle into a practical, learnable algorithm requires us to address three foundational challenges:

1. **The Nature of $U$**: What is the precise meaning of the latent variable $U$? How does it connect the observable features $X$ to the outcome $Y$?

2. **The Form of the Model**: How should we model the inference process $P(U|X)$ and the causal mechanism $f$? What distributional assumptions and functional forms are appropriate for causality?

3. **The Path to Tractability**: How can we combine these components into an end-to-end trainable system that is not only theoretically sound but also computationally efficient, avoiding the high costs of sampling-based inference?

The CausalEngine is our answer to these questions, built upon specific choices regarding the nature of $U$, the distribution for inference, and the structure of the causal mechanism.

### 3.1 THE DUAL ROLE OF U: SELECTION AND REPRESENTATION

The latent variable $U$ is the cornerstone of our framework. It embodies a dual role that bridges classical causal inference with modern representation learning, a concept formally established in (Gong et al., 2024). On one hand, $U$ acts as an **individual selection variable**, a population consists of heterogeneous individuals, and $U = u$ represents the complete set of attributes for a specific individual. These attributes determine how the individual would respond to any given treatment or condition. On the other hand, from a machine learning perspective, $U$ serves as an **individual causal representation**. It is a high-dimensional embedding that distills all causally relevant information from the input feature vector $X$. In contrast to standard representations that capture correlational patterns, $U$ is designed to capture only the factors that are causally determinative of the outcome $Y$. The objective of the first part of our model is therefore not merely to compress $X$, but to perform *abduction*: to infer the posterior probability distribution $P(U|X)$ that best explains the observed evidence.

Our formulation, $S = f(U, \varepsilon)$, directly separates two sources of randomness that were first unified in the mathematical framework of DiscoSCM (Gong et al., 2024). [2] The uncertainty regarding the individual selection variable $U$ represents the **epistemic uncertainty**, while the noise $\varepsilon$ represents the **aleatoric uncertainty** from the environment, which is the traditional focus of SCM. This explicit separation allows for a more granular and interpretable quantification of the total predictive uncertainty.

### 3.2 CORE TECHNICAL CHOICES FOR TRACTABLE LEARNING

To build a practical and efficient CausalEngine, we make two foundational design choices that distinguish our approach from conventional Deep Structural Causal Models (DSCMs) (Poinsot et al., 2024): the adoption of the Cauchy distribution for modeling uncertainty and the enforcement of a

---

[2]See Appendix C for detailed properties of DiscoSCM.

linear causal mechanism. These choices, while seemingly restrictive, work in concert to unlock a crucial property – fully analytical and tractable inference.

**The Language of Causality as Cauchy Distribution**   We posit that the choice of distribution for the latent cause $U$ is not a mere implementation detail but a fundamental modeling decision, grounded in the Potential Outcomes framework (Imbens & Rubin, 2015). This framework attributes outcome randomness to heterogeneous individuals, implying that when performing abduction—reasoning from evidence $X$ back to a cause $U$—a model must not foreclose possibilities by prematurely dismissing any individual, including "black swan" outliers. The ubiquitous, light-tailed Gaussian distribution violates this principle by rapidly dismissing events far from the mean. We therefore adopt the heavy-tailed Cauchy distribution, which satisfies this requirement by assigning non-negligible probability to even extreme individuals.

**The Assumption of Linear Causality**   Complementing our choice of distribution, we hypothesize that causal mechanisms are fundamentally simple. Inspired by the success of reward modeling in LLMs, where a simple linear function over a powerful representation can capture complex human preferences (Ouyang et al., 2022), we enforce a linear structure on the causal mechanism. The challenge is thus shifted from learning a complex, non-linear function $f$ to learning a powerful and expressive causal representation $U$ that linearizes the problem.

The synergy between the Cauchy distribution and the linear causal mechanism enables analytical uncertainty propagation throughout the model. For example, if $U \sim \text{Cauchy}(\mu_U, \gamma_U)$ and $S = W \cdot U + b$, then the output distribution is a Cauchy distribution with location parameter $\mu_S = W \cdot \mu_U + b$ and scale parameter $\gamma_S = |W| \cdot \gamma_U$. [3]

### 3.3 CAUSALENGINE FOUR-STAGE ARCHITECTURE

Having established the theoretical and technical foundations, we now detail the four-stage architecture that brings them to life: Perception, Abduction, Action, and Decision (Figure 1). We will now detail the four stages.

**Stage 1: Perception**   Extract a meaningful feature representation $Z$ from the raw input features $X$. This stage is analogous to the feature extractor or encoder in a standard deep learning model.

$$Z = \text{Perception}(X) \tag{2}$$

The key requirement for the perception module is that the resulting representation $Z$ must contain sufficient information to identify the underlying causal variable $U$. The specific implementation can be any suitable architecture, from a simple linear layer to a complex transformer, depending on the nature of the input data.

**Stage 2: Abduction**   Abduction is the process of reasoning from an observed effect $X$ to its most plausible latent cause $U$. Given the causal mechanism $g$, which is parameterized by a neural network, our goal is to find a distribution over $U$ that could have generated a given $X$. Formally, we aim to compute the posterior distribution $p(U|X)$. Since $g$ is a complex, non-linear function, an analytical solution for this posterior is generally intractable.

$$P(U|Z) = \text{Cauchy}(\mu_U(Z), \gamma_U(Z)) \tag{3}$$

We model this posterior as a Cauchy distribution. The location parameter $\mu_U$ represents the most likely individual causal representation, while the scale parameter $\gamma_U$ quantifies the *epistemic uncertainty*—our model's uncertainty about this inference. Both parameters are learned functions of the representation $Z$, typically implemented as linear heads:

$$\mu_U(Z) = W_{\text{loc}}Z + b_{\text{loc}} \tag{4}$$
$$\gamma_U(Z) = \text{softplus}(W_{\text{scale}}Z + b_{\text{scale}}) \tag{5}$$

The softplus function ensures that the scale parameter $\gamma_U$ is always positive.

---

[3]See Appendix B for detailed properties of Cauchy distribution.

**Stage 3: Action**    Model the causal mechanism that maps the cause $U$ to a decision score $S$. This stage represents the application of an invariant causal law. First, we introduce *aleatoric uncertainty* by injecting exogenous noise $\varepsilon$, which is also drawn from a standard Cauchy distribution. This noise represents irreducible environmental randomness.

$$U' = U + \varepsilon, \quad \text{where } \varepsilon \sim \text{Cauchy}(0, \mathbf{b}_{\text{noise}}) \tag{6}$$

Here, $\mathbf{b}_{\text{noise}}$ is a learnable parameter that controls the magnitude of the noise. Next, we apply the linear causal law to the noise-injected variable $U'$:

$$S = W_{\text{action}}U' + b_{\text{action}} \tag{7}$$

As explained previously, due to the linear stability of the Cauchy distribution, the distribution of the output score $S$ is also a Cauchy distribution, with its parameters $\mu_S$ and $\gamma_S$ computed analytically.

**Stage 4: Decision**    Transform the abstract decision score $S$ into a task-specific prediction $Y$.

$$Y = \tau(S) \tag{8}$$

The transformation $\tau$ depends on the task. For regression, we typically use an identity mapping, $\tau(s) = s$. The model is then trained by minimizing the negative log-likelihood of the true labels under the predicted Cauchy distribution for $S$:

$$\mathcal{L}_{\text{regression}} = -\log p_{\text{Cauchy}}(y_{\text{true}}|\mu_S, \gamma_S) = \log(\pi\gamma_S) + \log\left(1 + \left(\frac{y_{\text{true}} - \mu_S}{\gamma_S}\right)^2\right) \tag{9}$$

This loss function is robust by design, as its logarithmic growth for large errors naturally down-weights the influence of outliers, behaving similarly to a robust M-estimator.

## 4  EXPERIMENTS

This section presents a comprehensive empirical evaluation of CausalEngine[4] , structured around three key analyses: (1) robustness evaluation on synthetic data; (2) performance evaluation on public benchmarks; and (3) targeted ablation studies of its core components.

### 4.1  SYNTHETIC DATA: LABEL NOISE ROBUSTNESS

**Experimental Setup**    We conduct experiments on two synthetic datasets—**Statistical** and **Causal**—to evaluate robustness against four types of label noise: **Shuffle**, **Outlier**, **Asymmetric**, and **Systematic**. Detailed descriptions of the data generation processes and noise injection protocols are provided in Appendix A.1.1. We compare CausalEngine against a comprehensive suite of 13 models, including five neural network variants (PyTorch MLP, Pinball MLP, Cauchy MLP, Huber MLP, sklearn MLP) and four tree-based ensembles (XGBoost, LightGBM, CatBoost, Random Forest). To ensure a fair comparison, all baseline models were configured with optimized hyperparameters consistent with those used for CausalEngine, and these settings were kept constant across all experiments for a given model type. Our primary evaluation metric is the **Median Absolute Error (MdAE)**. Its reliance on the median provides a 50% breakdown point, making it exceptionally resilient to the extreme prediction errors introduced by label noise. Unlike mean-based metrics such as MAE and RMSE which are easily skewed by outliers, MdAE provides a stable assessment of a model's performance on the uncorrupted majority of the data, aligning perfectly with our experimental design. For a comprehensive assessment, we also report MAE, RMSE, and R-squared ($R^2$).

**Results and Analysis**    We first establish baseline performance in a noise-free (0%) environment. CausalEngine demonstrates competitive performance against the best baseline models, particularly on the Causal data for which it was designed (full results in Appendix A.1.3). This confirms that the causal architecture is effective without a prohibitive loss of accuracy in traditional statistical settings.

---

[4]The CausalEngine has different modes corresponding to different uncertainty modeling strategies. `Standard` is the default mode that incorporates both epistemic and aleatoric uncertainty, `Endogenous` uses solely on epistemic uncertainty $U$, and `Exogenous` considers only aleatoric uncertainty $\varepsilon$.

The advantages of CausalEngine become evident under high label noise. Under 40% shuffle noise, a condition that accentuates differences in model robustness, CausalEngine demonstrates a significant performance margin over all 13 baselines (detailed results in Appendix A.1.4). On the Statistical data, the best CE variant (MdAE 2.6860) achieves an **84.0%** error reduction compared to the strongest baseline, Pinball MLP (MdAE 16.7363). This advantage is maintained on the Causal data, where the top-performing CE variant (MdAE 0.2161) reduces the error by **55.7%** relative to the next-best model (MdAE 0.4873).

As visually confirmed in Figure 2, this performance advantage is consistent across all four tested noise types. These results support our core hypothesis: by modeling the underlying causal graph, the Causal Engine is more effective at disregarding spurious correlations introduced by label noise than traditional models that rely solely on statistical associations.

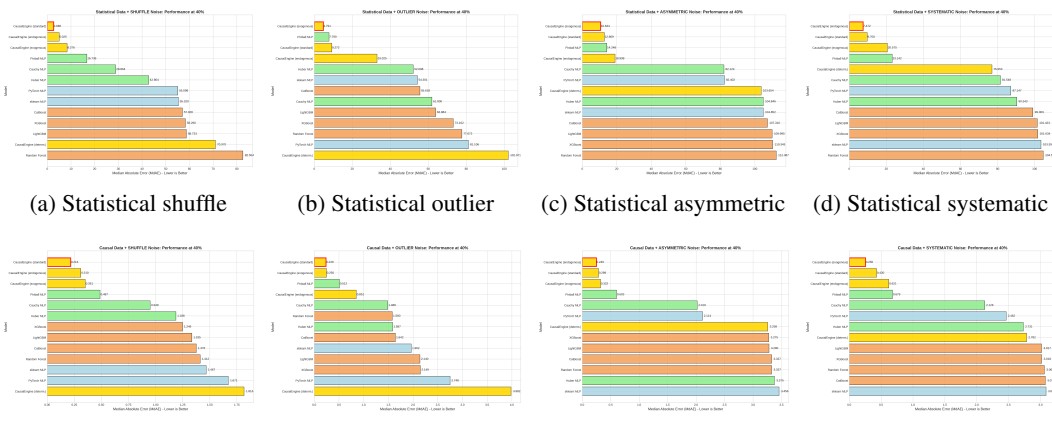

| (a) Statistical shuffle | (b) Statistical outlier | (c) Statistical asymmetric | (d) Statistical systematic |
| (e) Causal shuffle | (f) Causal outlier | (g) Causal asymmetric | (h) Causal systematic |

Figure 2: Robustness of CausalEngine vs. baselines on synthetic data (MdAE at 40% noise).

## 4.2 EVALUATION ON PUBLIC DATASETS

**Performance Across Datasets**  To assess its generalizability, we evaluated CausalEngine on eight public regression datasets spanning diverse domains (a summary is provided in Appendix A.2.1). In noise-free conditions, CausalEngine is competitive with strong baselines, achieving the top performance on 3 out of 8 datasets and confirming its efficacy for standard tasks (full results in Appendix A.2.2).

The key finding, however, is its robustness under label noise. As shown in Table 2, under a high-noise (40% shuffle noise) condition, the performance gains are substantial, with MdAE reduced by up to 3-fold on several datasets (e.g., bodyfat, space_ga). This pattern of superior performance is consistent across other standard metrics (MAE, RMSE, and R²), with full details available in the Appendix. This validates the adaptability and efficacy of our framework for real-world applications with imperfect data.

**Comparative Analysis Across Different Noise Types**
Our comparative analysis across four distinct noise types further confirms CausalEngine's superior robustness. Figure 3 illustrates this on the California dataset, where CausalEngine consistently outperforms baselines under high noise. This stability is further corroborated by the detailed robustness curves in Appendix A.2.3, which show CausalEngine's performance remaining stable while baselines degrade rapidly with increasing noise.

The performance advantage is especially pronounced under systematic and asymmetric noise. These noise types introduce consistent biases that confound traditional

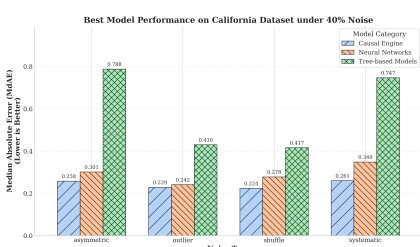

Figure 3: CausalEngine demonstrates robust performance on the California dataset across diverse noise types. The figure compares MdAE at a 40% noise level, showing that CausalEngine consistently achieves the lowest error against strong baselines.

Table 2: Performance of CausalEngine on public datasets under high noise (MdAE at 40% shuffle noise).

| Dataset | CE-Best | CE-Model | Best-Other | Other-Model | CE-Rank | Winner |
|---------|---------|----------|------------|-------------|---------|--------|
| concrete | 0.54 | Endogenous | 0.62 | Pinball | 1st | **CE-End** |
| space_ga | 0.046 | Exogenous | 0.12 | Pinball | 1st | **CE-Exo** |
| cpu_small | 1.68 | Endogenous | 1.03 | Pinball | 2nd | **Pinball** |
| tecator | 1.45 | Standard | 1.80 | Pinball | 1st | **CE-Std** |
| diamonds | 108 | Standard | 196 | Pinball | 1st | **CE-Std** |
| kin8nm | 0.055 | Standard | 0.067 | Pinball | 1st | **CE-Std** |
| bodyfat | 35.4 | Endogenous | 104 | Pinball | 1st | **CE-End** |
| california | 0.22 | Exogenous | 0.28 | Pinball | 1st | **CE-Exo** |

models reliant on statistical associations. In contrast, CausalEngine's focus on the underlying causal mechanism allows it to better isolate the true signal from such structured data corruption.

### 4.3 ABLATION STUDIES AND DESIGN ANALYSIS

**Validating the Contribution of Causal Components**   To isolate the contribution of the causal inference mechanism to model robustness, we conducted an ablation study comparing the full CausalEngine against a deterministic baseline. This baseline uses identical architecture but deactivates stochastic components by setting the scale parameter ($\gamma$) to zero, eliminating both endogenous and exogenous noise. This creates a controlled comparison that isolates the causal mechanism's impact on noise robustness. The results, presented in Table 3, reveal a clear performance difference. Under 40% label noise, the error of the "deterministic" mode surged by 94%–178%, whereas the causal modes constrained their error increase to a much lower range of 12%–81%. This contrast provides compelling evidence that the model's robustness is not an artifact of its architecture but a direct consequence of its core causal components.

Table 3: Ablation study of CausalEngine's causal components (MdAE and % error increase at 40% noise).

| Mode | Asymmetric | Outlier | Shuffle | Systematic |
|------|-----------|---------|---------|-----------|
| Standard | 0.317 (+55.6%) | 0.272 (+34.5%) | 0.241 (+20.1%) | 0.377 (+80.6%) |
| Endogenous | 0.275 (+46.9%) | 0.249 (+17.8%) | 0.226 (+11.8%) | 0.288 (+46.6%) |
| Exogenous | 0.258 (+33.4%) | 0.229 (+17.5%) | 0.224 (+13.7%) | 0.261 (+28.3%) |
| **Deterministic** | **0.645 (+177.7%)** | **0.510 (+124.3%)** | **0.437 (+94.6%)** | **0.581 (+154.5%)** |

**The Role of Distributional Choice in Robustness**   The distributional choice for the uncertainty terms ($U$ and $\varepsilon$) is a critical design decision for robustness. While the Normal distribution is commonly used, we hypothesized that the heavy-tailed Cauchy distribution would be more robust to label noise due to its reduced sensitivity to outliers. The results, visualized in Figure 4, confirm this hypothesis. Models using the Cauchy distribution consistently demonstrate substantially better performance and stability compared to their Normal distribution counterparts as noise levels increase. This result underscores that the heavy-tailed distributional assumption is an essential component of CausalEngine's robust design.

## 5   CONCLUSION AND DISCUSSION

In this paper, we introduced Causal Regression, a new framework that reframes traditional regression analysis through the lens of causality. Instead of modeling brittle statistical correlations, we focus on learning the invariant causal mechanisms that govern the data-generating process, summarized by

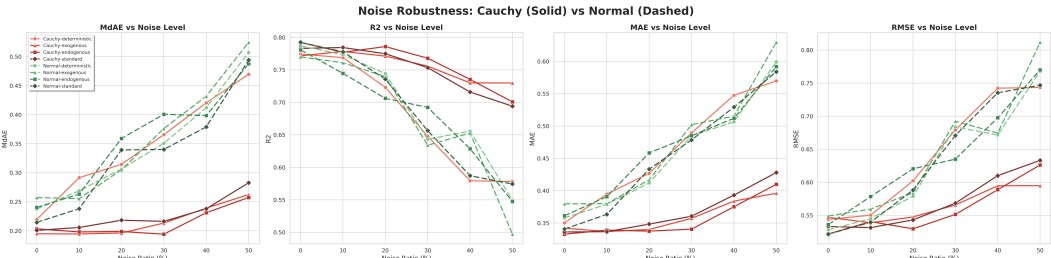

Figure 4: Impact of distributional choice on robustness: Cauchy vs. Normal distribution.

the structural equation $Y = f(U, \varepsilon)$. Our central argument is that a model's ability to achieve true robustness and interpretability hinges on its capacity to reason about underlying causes rather than surface-level associations.

To make this framework practical, we presented the CausalEngine, an interpretable-by-design four-stage architecture that operationalizes causal reasoning through Perception, Abduction, Action, and Decision. The cornerstone of CausalEngine is a novel analytical framework built on two deliberate technical choices: the Assumption of Linear Causality and the adoption of the Cauchy distribution as the language of causality. This synergy unlocks tractable, end-to-end learning by enabling the analytical propagation of uncertainty, thereby obviating the need for computationally expensive sampling-based inference. Our experiments demonstrated that this causal-first approach yields a significant boost in robustness against data label noise compared to established methods.

Looking forward, this work opens several promising avenues for future research.

- **Exploring Non-Linear Decision Functions:** The current framework utilizes a one-dimensional score $S$ to drive the final linear prediction. An important future direction is to explore non-linear decision functions from $S$ to $Y$ (e.g., by introducing non-linear activation functions). This would generalize CausalEngine to classification or more complex regression tasks, while also requiring investigation into how to maintain the model's tractability.

- **Analytical Inference from Scalar to Vector Scores:** To achieve tractable analytical inference, the current framework maps the high-dimensional latent cause $U$ to a one-dimensional scalar score $S$. A challenging extension is to explore elevating $S$ to a high-dimensional vector and to investigate if and how the sampling-free, analytical properties of our framework can be preserved in this more complex setting.

In conclusion, by shifting the objective of learning from fitting correlations to modeling causes, this work offers a path toward building machine learning systems that are not only accurate but also more reliable, transparent, and trustworthy in the face of real-world complexities.

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

## .1 ROBUST REGRESSION METHODS FOR PREDICTION.

A primary challenge in regression analysis is ensuring model performance remains stable when the data deviates from idealized assumptions, particularly in the presence of outliers. To address this, a rich literature on robust regression has developed. One major family of methods achieves robustness by redesigning the loss function to be less sensitive to large errors. M-estimators are a cornerstone of this approach, with the Huber loss being a canonical example that cleverly interpolates between a quadratic loss for small errors and an absolute loss for larger ones, thus mitigating the influence of outliers (Huber, 1964). Distinct from mean-based regression, Quantile Regression offers another powerful alternative by using an asymmetric pinball loss to model conditional quantiles of the outcome (Koenker & Bassett, 1978). This provides not only a more complete view of the response distribution but also inherent robustness.

A second family of methods derives robustness from their algorithmic structure rather than the loss function alone. Tree-based ensembles are a prime example, including Random Forests (Breiman, 2001) and Gradient Boosting Machines (Friedman, 2001). Their strength lies in the recursive partitioning of the feature space, which naturally isolates outlier observations into specific regions of the model, thereby limiting their global impact on the final prediction. However, despite their proven efficacy, these methods are designed for **statistical robustness**, aiming to produce reliable predictions that are insensitive to noise—a challenge extensively surveyed in the context of deep learning (**?**).

Their objective is not to model the underlying data-generating process itself. Consequently, they do not provide insights into the causal mechanisms that give rise to the data, a limitation that our work aims to address.

# A  ADDITIONAL EXPERIMENTS

## A.1  SYNTHETIC DATA EXPERIMENTS

### A.1.1  EXPERIMENTAL SETUP

To isolate and rigorously test CausalEngine's core robustness, we first evaluate it in a controlled synthetic environment where the ground-truth causal structures are known. This allows for a pure assessment of model capabilities, free from the confounding variables inherent in real-world data. We designed two distinct datasets:

**Statistical Data**: Generated following traditional regression assumptions:

$$y_i = \sum_{j=1}^{10} \beta_j x_{ij} + \epsilon_i \tag{10}$$

where $x_{ij} \sim \mathcal{N}(0,1)$, $\epsilon_i \sim \mathcal{N}(0, 0.1^2)$, with 10 effective features and 5 noise features.

**Causal Data**: Generated from a structural causal model to create a realistic inference challenge. The model includes unobserved confounders ($\mathbf{U}$), instrumental variables ($\mathbf{Z}$), a treatment ($T$), and a mediator ($M$). The final outcome $Y$ incorporates direct, mediation, confounding, heterogeneous, and nonlinear effects:

$$Y = 2.5T + 1.5M + 0.2U_1 + 0.1U_2 + 0.3X_{\text{conf},1} \cdot T + 0.2\sin(X_{\text{conf},2}) + \epsilon_Y \tag{11}$$

The model is trained on observable features while the true causal drivers $\mathbf{U}$ and $\mathbf{Z}$ remain hidden.

**Noise Injection Protocol**  We evaluate robustness against four distinct types of label noise, each designed to simulate a different real-world data corruption scenario. The noise types are:

- **Shuffle Noise**: A fraction of labels are randomly permuted, severing their true feature-label relationship to simulate severe annotation errors.
- **Outlier Noise**: A subset of labels are replaced by extreme values calculated as a multiple of the data's standard deviation from the mean, mimicking data entry mistakes or sensor malfunctions.
- **Asymmetric Noise**: A directional bias is introduced to a fraction of labels, where noise is added with a high probability (e.g., 90%) of being positive and a low probability of being negative. The magnitude of the noise is scaled by the standard deviation, simulating biased reporting.
- **Systematic Noise**: A constant offset is added to a subset of samples, representing a consistent sensor drift or a subgroup-specific systematic error.

### A.1.2  IMPLEMENTATION AND HYPERPARAMETER DETAILS

All experiments were conducted with a fixed random seed (42) for reproducibility. Datasets were split into training and testing sets with a 80/20 ratio. For all neural network-based models, including CausalEngine variants and MLP baselines, features were standardized using `StandardScaler`. Tree-based models were trained on the original, unscaled features. The specific hyperparameters for all models are detailed in Table 4 and Table 5.

### A.1.3  BASELINE PERFORMANCE (0% NOISE)

To establish a baseline, we first evaluated all models in a noise-free (0%) environment on the synthetic datasets. Table 6 provides a high-level summary of this comparison, showing the performance of the best CausalEngine variant against the best-performing baseline. For a comprehensive view,

Table 4: Hyperparameters for Neural Network Models (CausalEngine & MLPs)

| Parameter | Value |
|---|---|
| Hidden Layer Sizes | (128, 64, 32) |
| Max Epochs | 1500 |
| Learning Rate | 0.001 |
| Optimizer | Adam |
| Early Stopping Patience | 20 epochs |
| Early Stopping Tolerance | 1e-5 |
| Batch Size | 200 |
| L2 Regularization ($\alpha$) | 0.0001 |
| *CausalEngine-Specific* | |
| Gamma Init ($\gamma_{init}$) | 1.0 |
| b_noise Init ($b_{noise\_init}$) | 1.0 |
| b_noise Trainable | True |

Table 5: Hyperparameters for Tree Ensemble Models

| Parameter | Random Forest | XGBoost | LightGBM | CatBoost |
|---|---|---|---|---|
| n_estimators / iterations | 200 | 100 | 100 | 100 |
| max_depth / depth | 5 | 5 | 5 | 5 |
| learning_rate | N/A | 0.1 | 0.1 | 0.1 |
| random_state / random_seed | 42 | 42 | 42 | 42 |

Table 7 presents the detailed metrics for all 13 evaluated models. The results confirm that the causal architecture is highly effective in its target domain (causal systems) without a prohibitive loss of accuracy in traditional statistical settings.

Table 6: Performance comparison on synthetic datasets (0% noise)

| Data Type | CE MdAE | Best MdAE | CE R² | Best R² | MdAE Gap | R² Gap |
|---|---|---|---|---|---|---|
| Statistical | 1.4771 | 0.8654 | 0.999835 | 0.999943 | 0.6117 | -0.000108 |
| Causal | 0.1796 | 0.1760 | 0.997031 | 0.996531 | 0.0036 | 0.000500 |

Table 7: Full performance on synthetic datasets (0% noise, all models)

| Model | Statistical Data | | | | Causal Data | | | |
|---|---|---|---|---|---|---|---|---|
| | MAE | MdAE | MSE | R² | MAE | MdAE | MSE | R² |
| CE (standard) | 2.0217 | 1.7043 | 6.3781 | 0.9998 | 0.2323 | 0.1918 | 0.0857 | **0.9969** |
| CE (endogenous) | 4.5275 | 4.0958 | 30.0710 | 0.9990 | 0.3113 | 0.2586 | 0.1557 | 0.9943 |
| CE (exogenous) | 2.0721 | 1.7248 | 6.9270 | 0.9998 | 0.2343 | 0.1983 | 0.0859 | **0.9969** |
| CE (deterministic) | 3.6115 | 2.9257 | 21.7490 | 0.9993 | 0.3429 | 0.2878 | 0.1804 | 0.9934 |
| PyTorch MLP | **1.0363** | **0.8654** | **1.8182** | **0.9999** | 0.2388 | 0.1960 | 0.0904 | 0.9967 |
| Pinball MLP | 4.9503 | 3.2680 | 51.4964 | 0.9984 | 0.2487 | 0.1984 | 0.1043 | 0.9962 |
| Cauchy MLP | 5.7377 | 5.1400 | 50.9068 | 0.9984 | 0.2541 | 0.2037 | 0.1093 | 0.9960 |
| Huber MLP | 6.1204 | 5.3779 | 57.1158 | 0.9982 | 0.2741 | 0.2223 | 0.1265 | 0.9954 |
| sklearn MLP | 6.1219 | 4.7717 | 64.7157 | 0.9980 | 1.3411 | 1.1077 | 2.6288 | 0.9038 |
| XGBoost | 29.4874 | 21.9472 | 1596.44 | 0.9498 | **0.2259** | **0.1760** | **0.0948** | 0.9965 |
| LightGBM | 29.7608 | 23.5336 | 1584.17 | 0.9502 | 0.2449 | 0.1824 | 0.1458 | 0.9947 |
| CatBoost | 19.6447 | 14.0904 | 790.556 | 0.9752 | 0.3804 | 0.3013 | 0.2373 | 0.9913 |
| Random Forest | 84.3883 | 70.1995 | 11307.8 | 0.6446 | 1.5794 | 1.2829 | 3.7380 | 0.8631 |

### A.1.4 PERFORMANCE UNDER HIGH NOISE (40%)

Table 8 presents the complete performance metrics for all 13 models evaluated on the synthetic datasets under a high-noise condition (40% shuffle noise). This table provides the detailed data supporting the summary analysis in the main text.

Table 8: Performance on synthetic datasets under 40% shuffle noise

| Model | Statistical Data | | | | Causal Data | | | |
|---|---|---|---|---|---|---|---|---|
| | MAE | MdAE | RMSE | R² | MAE | MdAE | RMSE | R² |
| CE (standard) | **3.3687** | **2.6860** | **4.4949** | **0.9994** | **0.2680** | **0.2161** | **0.3414** | **0.9957** |
| CE (endogenous) | 6.0928 | 5.0251 | 7.8865 | 0.9980 | 0.3769 | 0.3105 | 0.4851 | 0.9914 |
| CE (exogenous) | 10.2698 | 8.3755 | 13.3199 | 0.9944 | 0.4339 | 0.3508 | 0.5576 | 0.9886 |
| CE (deterministic) | 85.2951 | 70.9702 | 109.0554 | 0.6262 | 2.3431 | 1.8158 | 3.0415 | 0.6614 |
| PyTorch MLP | 66.9778 | 55.0978 | 84.8953 | 0.7735 | 2.1277 | 1.6706 | 2.7673 | 0.7197 |
| Pinball MLP | 21.1034 | 16.7363 | 27.4688 | 0.9763 | 0.6408 | 0.4873 | 0.8453 | 0.9738 |
| Cauchy MLP | 33.7622 | 28.8644 | 41.6595 | 0.9454 | 1.1873 | 0.9494 | 1.5268 | 0.9147 |
| Huber MLP | 49.9472 | 42.8041 | 62.5510 | 0.8770 | 1.4981 | 1.1875 | 1.9427 | 0.8619 |
| sklearn MLP | 65.3076 | 55.3204 | 82.9221 | 0.7839 | 1.8438 | 1.4671 | 2.4336 | 0.7832 |
| XGBoost | 72.2841 | 58.2897 | 92.4185 | 0.7315 | 1.8051 | 1.2490 | 2.5678 | 0.7587 |
| LightGBM | 71.4519 | 58.7327 | 90.9935 | 0.7397 | 1.7743 | 1.3348 | 2.4291 | 0.7841 |
| CatBoost | 69.0617 | 57.0279 | 88.8874 | 0.7516 | 1.7307 | 1.3758 | 2.3411 | 0.7994 |
| Random Forest | 99.6165 | 82.5539 | 126.4897 | 0.4971 | 1.7495 | 1.4119 | 2.3528 | 0.7974 |

## A.2 PUBLIC DATASET EXPERIMENTS

### A.2.1 DATASET SUMMARY

To assess the generalizability and practical applicability of CausalEngine, we conducted evaluations across a curated subset of eight public regression datasets, selected to represent a diversity of sample sizes, feature dimensions, and domains. Table 9 provides a summary of these datasets.

Table 9: Summary of public datasets. Each column represents a dataset, showing its number of features and samples.

| Property | concrete | space_ga | cpu_small | tecator | diamonds | kin8nm | bodyfat | california |
|---|---|---|---|---|---|---|---|---|
| Features | 8 | 48 | 14 | 124 | 9 | 8 | 3 | 8 |
| Samples | 1,030 | 15,000 | 252 | 240 | 53,940 | 8,192 | 475 | 20,640 |

### A.2.2 BASELINE PERFORMANCE (0% NOISE)

This section details the baseline performance evaluation on the eight public datasets under noise-free conditions. Table 10 summarizes the results, listing the best-performing model for each dataset alongside key metrics (MdAE, MAE, RMSE, and R²). These findings establish that CausalEngine is competitive with strong baseline models in a standard setting, achieving the top performance on 3 out of 8 datasets. This confirms its efficacy for standard regression tasks and provides a solid foundation before evaluating its core contribution—robustness to label noise.

### A.2.3 DETAILED ROBUSTNESS CURVES

Figure 5 presents the detailed robustness curves for the California dataset across all four noise types. Each plot shows the degradation of four key metrics (MdAE, R², MAE, RMSE) as the noise ratio increases from 0% to 50%. These plots provide a granular view of CausalEngine's consistent performance advantage over baseline models across the full spectrum of noise intensities.

Table 10: Baseline performance comparison on public datasets under noise-free (0%) conditions. For each dataset, the best performing model is shown along with its key metrics.

| Dataset | Best Model | MdAE | MAE | RMSE | R² |
|---|---|---|---|---|---|
| concrete_strength | Random Forest | 0.0000 | 0.0228 | 0.0670 | 0.9997 |
| space_ga | **CausalEngine** | **0.1314** | **1.1273** | **3.0450** | **0.9945** |
| cpu_small | Random Forest | 0.1363 | 0.2014 | 0.2815 | 0.9983 |
| tecator | XGBoost | 0.5056 | 0.8865 | 1.4559 | 0.9901 |
| diamonds | **CausalEngine** | **97.0655** | **274.4781** | **575.5897** | **0.9792** |
| kin8nm | Cauchy MLP | 0.0436 | 0.0551 | 0.0715 | 0.9262 |
| bodyfat | Random Forest | 73.5307 | 375.6143 | 952.6123 | 0.9410 |
| california | **CausalEngine** | **0.1837** | **0.3276** | **0.5283** | **0.7870** |

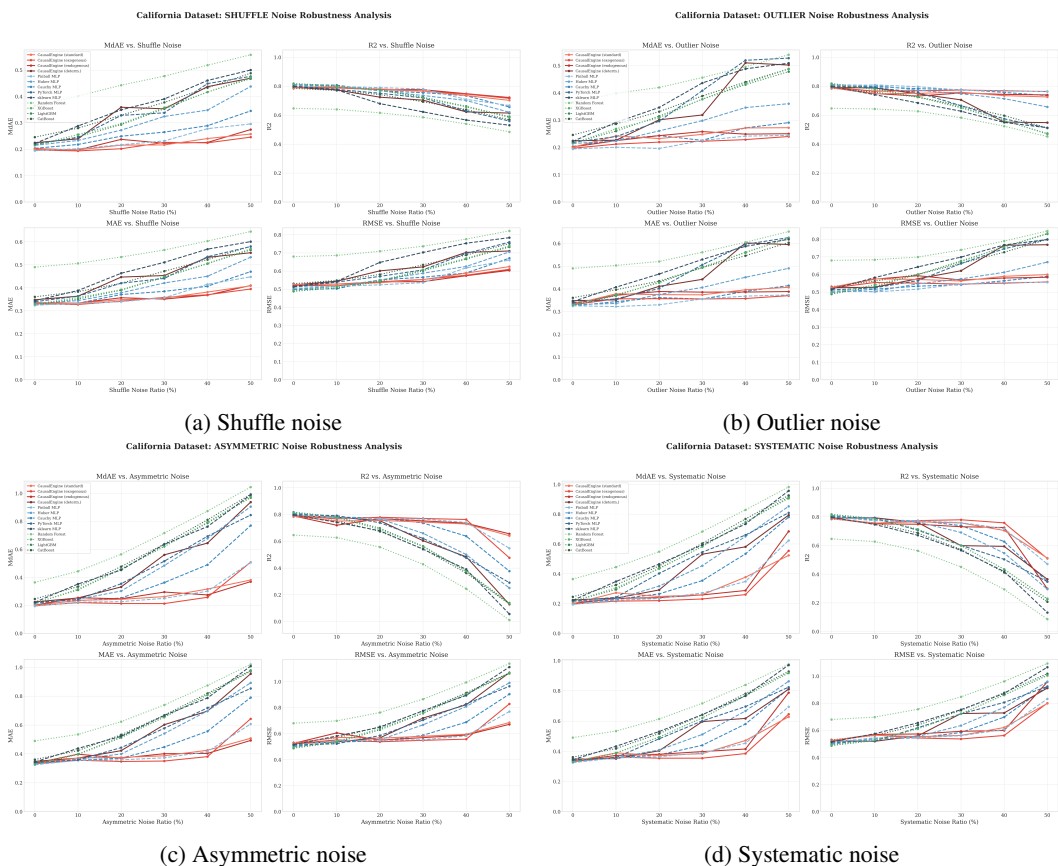

(a) Shuffle noise        (b) Outlier noise

(c) Asymmetric noise        (d) Systematic noise

Figure 5: Detailed robustness curves on the California dataset. Performance metrics are evaluated against increasing levels of four different noise types.

## B  PROPERTIES OF THE CAUCHY DISTRIBUTION

The Cauchy distribution, centered at $\mu$ with a scale parameter $\gamma$, is a continuous probability distribution with several key mathematical properties relevant to our framework.

The probability density function (PDF) is given by:

$$p(x; \mu, \gamma) = \frac{1}{\pi\gamma\left[1 + \left(\frac{x-\mu}{\gamma}\right)^2\right]}$$

- **Heavy Tails:** The tails of the distribution decay polynomially, meaning it assigns significantly higher probability to extreme values compared to a Gaussian distribution.

- **Undefined Moments:** The mean, variance, and higher moments of the Cauchy distribution are undefined.

- **Linear Stability (Additivity):** The Cauchy distribution is a stable distribution. A linear combination of independent Cauchy-distributed random variables is also Cauchy-distributed. If $X_1 \sim \text{Cauchy}(\mu_1, \gamma_1)$ and $X_2 \sim \text{Cauchy}(\mu_2, \gamma_2)$, then for any real constants $a$ and $b$:

$$aX_1 + bX_2 \sim \text{Cauchy}(a\mu_1 + b\mu_2, |a|\gamma_1 + |b|\gamma_2)$$

A crucial property for machine learning applications is that the KL divergence between two Cauchy distributions has a closed-form analytical expression (**?**). For two Cauchy distributions, $P_1 = \text{Cauchy}(\mu_1, \gamma_1)$ and $P_2 = \text{Cauchy}(\mu_2, \gamma_2)$, the KL divergence is:

$$D_{KL}(P_1||P_2) = \log\left(\frac{(\gamma_1 + \gamma_2)^2 + (\mu_1 - \mu_2)^2}{4\gamma_1\gamma_2}\right)$$

This property allows for efficient computation of the divergence, making it a promising regularization term in the overall loss function, without resorting to numerical estimation.

## C  CAUSAL MODELING FRAMEWORKS

The principal frameworks in causal modeling, Potential Outcomes (PO) (Neyman, 1923; Rubin, 1974; Imbens & Rubin, 2015) and Structural Causal Models (SCM) (Pearl, 2009), hold theoretical equivalence and are both anchored in the consistency rule (Angrist et al., 1996; Cole & Frangakis, 2009; Pearl, 2010). The PO approach centers on experimental units, emphasizing individual semantics, whereas SCM is based on structural equations, from which it delineates three levels of causal information: associational, interventional, and counterfactual.

The PO framework, also known as the Rubin Causal Model (Holland, 1986), begins with a population of experiment units. There is a treatment that can take on different values for each unit. Each unit in the population is characterized by a set of potential outcomes $Y(t)$, one for each value of the treatment. Only one of these potential outcomes can be observed, namely the one corresponding to the treatment received:

$$Y = \sum_t Y(t)\mathbf{1}_{T=t}. \tag{12}$$

This equation is a derivation of the consistency assumption.

**Assumption 1** (**Consistency**)**.** *The potential outcome $Y(t)$ precisely matches the observed variable $Y$ given observed treatment $T = t$, i.e.,*

$$T = t \Rightarrow Y(t) = Y. \tag{13}$$

The framework of *structural causal models* (SCMs) is presented as follows.

**Definition 1** (**Structural Causal Models** (Pearl, 2009))**.** *An SCM is a tuple $\langle \mathbf{U}, \mathbf{V}, \mathcal{F} \rangle$, where*

- $\mathbf{U}$ *is a set of background variables, also called exogenous variables, that are determined by factors outside the model, and $P(\cdot)$ is a probability function defined over the domain of $\mathbf{U}$;*

- $\mathbf{V}$ *is a set $\{V_1, V_2, \ldots, V_n\}$ of (endogenous) variables of interest that are determined by other variables in the model – that is, in $\mathbf{U} \cup \mathbf{V}$;*

- $\mathcal{F}$ *is a set of functions $\{f_1, f_2, \ldots, f_n\}$ such that each $f_i$ is a mapping from (the respective domains of) $U_i \cup Pa_i$ to $V_i$, where $U_i \subseteq \mathbf{U}$, $Pa_i \subseteq \mathbf{V} \setminus V_i$, and the entire set $\mathcal{F}$ forms a mapping from $\mathbf{U}$ to $\mathbf{V}$. That is, for $i = 1, \ldots, n$, each $f_i \in \mathcal{F}$ is such that*

$$v_i \leftarrow f_i(pa_i, u_i),$$

*i.e., it assigns a value to $V_i$ that depends on (the values of) a select set of variables in $\mathbf{U} \cup \mathbf{V}$.*

Interventions are defined through a mathematical operator.

**Definition 2** (**Interventional SCM** (Pearl, 2009)). *Consider an SCM $\langle \mathbf{U}, \mathbf{V}, \mathcal{F} \rangle$, with a set of variables $\mathbf{X}$ in $\mathbf{V}$, and a particular realization $\mathbf{x}$ of $\mathbf{X}$. The $do(\mathbf{x})$ operator, representing an intervention (or action), modifies the set of structural equations $\mathcal{F}$ to $\mathcal{F}_{\mathbf{x}} := \{f_{V_i} : V_i \in \mathbf{V} \setminus \mathbf{X}\} \cup \{f_X \leftarrow x : X \in \mathbf{X}\}$ while maintaining all other elements constant. Consequently, the induced tuple $\langle \mathbf{U}, \mathbf{V}, \mathcal{F}_{\mathbf{x}} \rangle$ is called as Interventional SCM, and potential outcome $\mathbf{Y}(\mathbf{x})$ (or denoted as $\mathbf{Y}_{\mathbf{x}}(\mathbf{u})$) is defined as the set of variables $\mathbf{Y} \subseteq \mathbf{V}$ in this submodel.*

Formally, an SCM gives valuation for associational, interventional and counterfactual quantities in the Pearl Causal Hierarchy (PCH) as follows.

**Definition 3** (**Layer Valuation** (Bareinboim et al., 2022)). *An SCM $\langle \mathbf{U}, \mathbf{V}, \mathcal{F} \rangle$ induces a family of joint distributions over potential outcomes $\mathbf{Y}(\mathbf{x}), \ldots, \mathbf{Z}(\mathbf{w})$, for any $\mathbf{Y}, \mathbf{Z}, \ldots, \mathbf{X}, \mathbf{W} \subseteq \mathbf{V}$:*

$$P(\mathbf{y}_{\mathbf{x}}, \ldots, \mathbf{z}_{\mathbf{w}}) = \sum_{\{\mathbf{u} \mid \mathbf{Y}(\mathbf{x}) = \mathbf{y}, \ldots, \mathbf{Z}(\mathbf{w}) = \mathbf{z}\}} P(\mathbf{u}). \tag{14}$$

*is referred to as Layer 3 valuation. In the specific case involving only one intervention, e.g., $do(\mathbf{x})$:*

$$P(\mathbf{y}_{\mathbf{x}}) = \sum_{\{\mathbf{u} \mid \mathbf{Y}(\mathbf{x}) = \mathbf{y}\}} P(\mathbf{u}), \tag{15}$$

*is referred to as Layer 2 valuation. The even more specialized case when $\mathbf{X}$ is empty:*

$$P(\mathbf{y}) = \sum_{\{\mathbf{u} \mid \mathbf{Y} = \mathbf{y}\}} P(\mathbf{u}). \tag{16}$$

*is referred to as Layer 1 valuation. Here, $\mathbf{y}$ and $\mathbf{z}$ represent the observed outcomes, $\mathbf{x}$ and $\mathbf{w}$ the observed treatments, $\mathbf{u}$ the noise instantiation, and we denote $\mathbf{y}_{\mathbf{x}}$ and $\mathbf{z}_{\mathbf{w}}$ as the realization of their corresponding potential outcomes.*

In the case of recursive SCMs, the $do$-calculus can be employed to completely identify all Layer 2 expressions (Pearl, 1995; Huang & Valtorta, 2012). However, calculating counterfactuals at Layer 3 is generally far more challenging compared to Layers 1 and 2. This is because it essentially requires modeling the joint distribution of potential outcomes, such as the potential outcomes with and without aspirin. Unfortunately, we often lack access to the underlying causal mechanisms and only have observed traces of them. This limitation leads to the practical use of Eq. equation 14 for computing counterfactuals being quite restricted (Pearl, 2009).

**Assumption 2** (**Distribution-consistency** (Gong et al., 2024)). *For any individual represented by $U = u$ with an observed treatment $X = x$, the counterfactual outcome $Y(x)$ is equivalent in distribution to the observed outcome $Y$. Formally,*

$$X = x, U = u \Rightarrow Y(x) \stackrel{d}{=} Y, \tag{17}$$

*where $\stackrel{d}{=}$ indicates equivalence in distribution.*

Differing from Assumption 13, key modifications include the inclusion of $U = u$ and the use of $\stackrel{d}{=}$. In alignment with these modifications, the Distribution-consistency Structural Causal Model (DiscoSCM) (Gong et al., 2024) is proposed:

**Definition 4.** *A DiscoSCM is a tuple $\langle U, \mathbf{E}, \mathbf{V}, \mathcal{F} \rangle$, where*

- *$U$ is a unit selection variable, where each instantiation $U = u$ denotes an individual. It is associated with a probability function $P(u)$, uniformly distributed by default.*

- *$\mathbf{E}$ is a set of exogenous variables, also called noise variables, determined by factors outside the model. It is independent to $U$ and associated with a probability function $P(\mathbf{E})$;*

- *$\mathbf{V}$ is a set of endogenous variables of interest $\{V_1, V_2, \ldots, V_n\}$, determined by other variables in $\mathbf{E} \cup \mathbf{V}$;*

- *$\mathcal{F}$ is a set of functions,*

$$\{f_1(\cdot, \cdot; u), f_2(\cdot, \cdot; u), \ldots, f_n(\cdot, \cdot; u)\},$$

  *where each $f_i$ is a mapping from $E_i \cup Pa_i$ to $V_i$, with $E_i \subseteq \mathbf{E}$, $Pa_i \subseteq \mathbf{V} \setminus V_i$, for individual $U = u$. Each function assigns a value to $V_i$ based on a selected set of variables in $\mathbf{E} \cup \mathbf{V}$. That is, for $i = 1, \ldots, n$, each $f_i(\cdot, \cdot; u) \in \mathcal{F}$ is such that*

$$v_i \leftarrow f_i(pa_i, e_i; u).$$

Intervention and counterfactual logic within the DiscoSCM framework can be similarly defined. This framework introduces a novel lens – individual/population – to address causal questions. Specifically, consider a DiscoSCM where $e$ represents the observed trace or evidence (e.g., $X = x, Y = y$), we have:

**Theorem 1** (**Individual-Level Valuations** (Gong et al., 2024))**.** *For any given individual $u$,*

$$P(y_x|e; u) = P(y_x; u) = P(y|x; u)$$

*indicating that the (individual-level) probabilities of an outcome at Layer 1/2/3 are equal.*

**Theorem 2** (**Population-Level Valuations** (Gong et al., 2024))**.** *Consider a DiscoSCM wherein $Y(x)$ is the counterfactual outcome, and $e$ represents the observed trace or evidence. The Layer 3 valuation $P(Y^d(x)|e)$ is computed through the following process:*

***Step 1 (Abduction):*** *Derive the posterior distribution $P(u|e)$ of the unit selection variable $U$ based on the evidence $e$.*

***Step 2 (Valuation):*** *Compute individual-level valuation $P(y_x; u)$ for each unit $u$.*

***Step 3 (Reduction):*** *Aggregate these individual-level valuations to obtain the population-level valuation as follows:*

$$P(Y^d(x) = y|e) = \sum_u P(y_x; u)P(u|e), \tag{18}$$

## D  LANGUAGE AND LITERATURE SUPPORT

In the preparation of this manuscript, Large Language Models (LLMs) were utilized as support tools to enhance the quality and rigor of the text. The specific applications of LLMs were as follows:

1. **Grammar and Spelling Correction:** LLMs were used for detailed grammar and spelling checks throughout the document to ensure linguistic accuracy and professionalism.

2. **Sentence Refinement and Optimization:** For complex or awkward sentences, LLMs provided alternative phrasing suggestions, from which the clearest and most fluent expressions were selected to improve readability.

3. **Literature Search Assistance:** During the literature review phase, LLMs served as advanced search engines to assist in quickly locating and summarizing prior work in relevant fields, ensuring the comprehensiveness and timeliness of our research.

It is important to emphasize that LLMs served solely as auxiliary tools to improve writing efficiency and quality. All core ideas, experimental designs, result analyses, and final conclusions presented in this paper were independently conceived and executed by the authors.

