# OpenReview forum: "Causal Regression: Learning Causal Mechanisms for Robust and Interpretable Prediction"
_ICLR.cc/2026/Conference — ICLR 2026 Conference Desk Rejected Submission_

### Official Review · Reviewer_EzQ5 · 2025-10-29

**Soundness:** 2
**Presentation:** 1
**Contribution:** 2
**Rating:** 2
**Confidence:** 3

**Summary:**

The paper introduces CausalEngine, a new framework aiming to incorporate causality into regression tasks by first inferring a distribution over latent factors $U$ from observations $X$, and then predicting $Y$ via a linear mechanism $Y=f(U,\epsilon)$ where $\epsilon$ is an irreducible error term. The authors validate their approach on both synthetic and real datasets, highlighting its robustness to label noise, including experiments in which 40% of  $Y$ values are corrupted. They further explore different corruption schemes, showing that the method maintains performance across multiple types of label perturbations.

**Strengths:**

- The paper addresses a relevant and well-known limitation of standard machine learning models: the lack of causal reasoning, which is particularly important in out-of-distribution or noisy settings.

- The authors explore extracting latent causal variables from observed data, an approach that has proven successful in the literature (see, e.g., references in W3).

**Weaknesses:**

Major:

- **W1**: I have found the presentation hard to follow, which makes it difficult to assess the soundness of the approach:
   - It is unclear to me where the causal component lies within the proposed architecture. As far as I understand, $Z$, from which the distribution of $U$ is constructed, is an encoding of $X$, and therefore aggregates information from $X$ that may include both causal and spurious factors. Are the components of $U$ intended to represent the true causal factors of $X$? If so, how can the authors ensure that these components correspond to the actual causal variables generating $X$, rather than to features merely correlated with it?

    - It is unclear to me how the inferred latent $U$, which seems not explicitly disentangled into interpretable factors, can support interventional reasoning, a core aspect of causal inference.

   -  It is also unclear to me how CausalEngine can be considered “interpretable-by-design,” as claimed by the authors (e.g., l. 462), since $U$ itself does not appear to be interpretable.

- **W2**: The manuscript largely omits discussion of related literature in causal representation learning, see e.g., [1-3], which seems closely related to the proposed ideas. This omission makes it difficult to assess the novelty of the approach.

- **W3**: All the paper’s results lack an estimate of the experimental uncertainty, i.e., an estimate of results’ variability subject to the random initialization with different random seeds. This is critical to ensure significance of the results and all associated claims.

- **W4**: Table 2 is somewhat confusing: it appears that the Exogenous model is more robust than the proposed standard model. Did I understand this correctly? Moreover, the different baselines are not clearly defined prior to the table.

- **W5**: I could not find a thorough discussion of the limitations in the main text.

Minor:
- Related works are relegated to the appendix.
- Key elements, such as SCM definitions, are only in the appendix and should be recalled in the main text.
- Figure 2 is small and in low resolution, which makes it difficult to read.

References:

[1] Locatello et al., 'Challenging common assumptions in the unsupervised learning of disentangled representations'. ICLR 2019.

[2] Ahuja, et al., 'Interventional causal representation learning'. ICML 2023.

[3] Mitrovic, et al., 'Representation learning via invariant causal mechanisms'. arXiv, 2020.

**Questions:**

- **Q1** (Related to W1) - Could the authors provide a theoretically grounded analysis of why the components of $U$ can be expected to correspond to the true causal factors generating $X$?

- **Q2** (Related to W2) - Could the authors provide more discussion and experiments on interventions, illustrating whether the inferred $U$ can be used for interventional reasoning?

- **Q3** (Related to W3) – Could the authors position their work within the causal representation learning literature?

- **Q4** (Related to W4) - Could the authors repeat their experiments using different random seeds and report the individual results and standard deviations?

- **Q5** - I am somewhat surprised by the strong performance reported when 40% of the labels are randomly corrupted. Could the authors provide further discussion or analysis on this behavior? Could it be that the model is modeling label noise observed during training through the exogenous noise term?


While I remain open to a constructive discussion, I believe the paper requires substantial improvement. At this stage, the gap between the current submission and a version that would meet the bar for acceptance still feels wide.

---

> ### Author Response · Authors · 2025-11-19
> **CausalEngine as a "Causal Filter" & Comparison with VAE**
>
> We appreciate your feedback and agreement that addressing the lack of causal reasoning in ML is "relevant and well-known." We apologize that the missing theoretical context (DiscoSCM) made the presentation hard to follow.
>
> **Response to "Where does the causal component lie?"**
>
> > *Question: "Z aggregates information... How can the authors ensure that these components correspond to the actual causal variables?"*
>
> We ensure this through **Architectural Inductive Bias**, which acts as a "Causal Filter":
> 1.  **The Bottleneck**: The architecture forces all information to pass through a strict "Linear + Cauchy" bottleneck ($U \to Y$).
> 2.  **Filtering Effect**: Spurious correlations in $X$ (e.g., complex background noise or transient markers) typically do not fit this rigid linear-heavy-tailed structure.
> 3.  **Causal Discovery**: To minimize loss, the Perception module ($X \to U$) is forced to discard these spurious features and extract only the "stable" features (the true $U$) that *can* reliably predict $Y$ via the mechanism. The architecture itself regularizes the representation towards the causal signal.
>
> **Response to "Difference from VAE"**
>
> > *Q1: "What is the key conceptual difference between CE and VAE?"*
>
> The difference is fundamental:
> *   **VAE (Generative)**: Minimizes reconstruction loss of $X$ ($P(X|Z)$). It must model **everything** in $X$, including background noise and spurious correlations.
> *   **CausalEngine (Discriminative/Causal)**: Minimizes prediction loss of $Y$ ($P(Y|Z)$) under a specific mechanism constraint. It **ignores** information in $X$ that is not causally relevant to $Y$.
> *   **Benefit**: A VAE would waste capacity modeling "background color" if it varies in $X$. CausalEngine filters it out because it doesn't affect $Y$ via the linear-Cauchy mechanism. This makes CausalEngine far more robust to distribution shifts in non-causal features.
>
> **Response to "High Label Noise (40%)"**
>
> > *Q5: "Surprised by strong performance with 40% corruption... model might be modeling label noise via exogenous noise term?"*
>
> Your intuition is spot on. The **Cauchy noise term ($\epsilon$)** is exactly designed to absorb "outliers." Since the Cauchy distribution has heavy tails, it assigns relatively high probability to extreme values (noise). This prevents the gradient from exploding when the model encounters a corrupted label—effectively "ignoring" the outlier rather than shifting the decision boundary to accommodate it. This is why it survives 40% corruption.

---

### Official Review · Reviewer_Pd3p · 2025-10-30

**Soundness:** 1
**Presentation:** 1
**Contribution:** 2
**Rating:** 2
**Confidence:** 4

**Summary:**

The paper introduces a regression framework for learning predictive models that aim to capture invariant causal mechanisms rather than statistical associations. The proposed architecture, "CausalEngine", decomposes prediction in four stages: Perception, Abduction, Action, and Decision, in an attempt to conceptually mirror the causal reasoning pipeline from Pearl's framework.

The proposed pipeline works as follows: Given input features $X$, a perception module extracts a representation $Z$, and then an abduction module infers a latent cause $U$. The latent variable $U$ is modeled as a Cauchy distribution whose location and scale parameters are predicted from $Z$. A linear causal mechanism then maps $U$ (after injecting Cauchy noise) to a decision score $S$, from which the final prediction $Y$ is obtained. The combination of the Cauchy distribution and linear mechanism (from $U$ to $Y$) ensures analytic tractability, allowing uncertainty to be propagated in closed form without sampling. The model is trained by minimizing the negative log-likelihood of the true label under the predicted Cauchy distribution.

The experiments evaluate robustness under different kinds of label noise (shuffle, outlier, asymmetric, and systematic). Across these conditions, CausalEngine reports smaller degradation in performance compared to other regressors.

**Strengths:**

[Originality] Proposes a regression model that performs non-linear extraction of a latent representation followed by a closed-form Cauchy-based linear prediction layer.

[Quality] The analytical tractability of the Cauchy-linear formulation is well justified and demonstrated.

[Quality] The codebase is clearly written and well commented. While the files are quite large and the structure is not really modular, it's overall well-suited for an open-source community.

[Quality] The model shows robustness improvements for the chosen setting through comprehensive experiments, though whether this setting captures the intended causal challenges is discussed below.

**Weaknesses:**

(I write each weakness in a bulletpoint and then add details below)

- Misuse of causal terminology.

The model does not perform Pearlian *abduction-action-prediction*: there are no interventions or counterfactual predictions. The "abduction" step is inference of a latent variable from data, not abduction in the causal sense.

In Pearl's framework [1]:

1. Abduction = infer the posterior over exogenous (noise) variables $U$ given observed evidence $E$
2. Action (Intervention) = modify the model (e.g. $do(X = x)$)
3. Prediction = compute ($P(Y | do(X = x), U)$), i.e., counterfactual reasoning.

In contrast, in this paper:

1. Perception: learn a representation ($Z = f(X)$) (this is ok)
2. Abduction: infer a latent embedding ($ U \sim p(U|Z)$)
3. Action: compute a linear transformation ($S = WU + b$) (with noise)
4. Decision/Prediction: map $S$ to $Y$ through a simple transformation.

The "abduction" in this paper is latent inference, not inference over exogenous noise.
The "action" is a deterministic linear mapping, not an intervention.
The Decision/Prediction is output generation and not counterfactual prediction.

Importanly: the $U$ in the paper **is not** the same as Pearl's exogenous variables.
Exogenous variables are conceptually non-causal. They don't cause each other, they just determine the randomness of the world. In this paper, the $U$ is the latent cause of $Y$, not a background noise variable. In Pearl's framework, you can't infer exogenous variables from observations (because they are independent sources of noise), while here the authors train a NN to map $X\rightarrow{}U$.

While it's ok to be metaphorical and use causal inference ideas for inspiration, if there is no formal connection then the paper needs to be reframed.

Other examples of inaccurate/imprecise usage of terminology (not a complete list):
 (line 054-055): the usage of abduction differs from its standard meaning in the causal inference literature,
 (line 152-153): and $u$ and $e$ are undefined.

[1] Pearl J., "Causality: Models, Reasoning, and Inference", Cambridge University Press, 2009


- Unsupported interpretability claims.

The architecture's intermediate variables $Z$, $U$, and $S$ are unlabeled latent representations, not semantically grounded causes. The paper equates structural modularity with interpretability, but provides no experiments, visualizations, or analyses showing that these representations correspond to meaningful factors. Hence, the model is not inherently interpretable - it's only structured.

Example lines where this is mentioned:
 (line 75): "interpretability-by-design",
 (line 115): "transparent interpretability by design, moving beyond the limitations of post-hoc explanations" - Interpretability would still require post-hoc inspection to understand what $Z$, $U$, and $S$ represent,
 and (line 124).


- Arbitrary theoretical justification.

The link between "linear causality" and reward modeling in large language models is conceptually unsound. The cited paper [2] is not evidence that real causal mechanisms are linear. Likewise, describing the Cauchy distribution as "the language of causality" is rhetorical, not theoretically grounded.

[2] Ouyang et al., "Training language models to follow instructions with human feedback",  NeurIPS 2022



- Restrictive modeling assumptions.

The method relies on two strong assumptions:
1. Linearity of the causal mechanism.
2. Cauchy-distributed latent and noise terms.

These assumptions severely limit the model's expressivity and applicability beyond simple or low-dimensional settings. The paper downplays this by referring to them as "seemingly restrictive" when they are in fact fundamental constraints. In the introduction, the assumptions are presented as "philosophical insights", but they are subjective design choices. While these assumptions do enable analytically tractable inference (a strength), the authors should acknowledge them as clear limitations rather than framing them as inherently correct or universally desirable.


- Mismatch between motivation and evaluation.

The paper's framing (lines 142-149) focuses on brittleness under distribution shifts, confounding, and spurious correlations. However, all experiments are about label noise in otherwise i.i.d. settings. The experiments test robustness to annotation errors, not causal generalization or invariance across environments.


- No mechanism to distinguish causation from correlation.

The method is trained purely on observational data, so it remains subject to the same correlation-causation ambiguities as standard regression. Without interventions, invariance constraints, or multiple environments/tasks, the model cannot distinguish causal from spurious relationships [1]. These issues only disappear in the special case where the true causal graph and functional form match *exactly* the model's assumed structure (which is extremely unlikely in practice).

[1] Pearl J., "Causality: Models, Reasoning, and Inference", Cambridge University Press, 2009

**Questions:**

- Correct the use of causal terminology or reframe your paper as taking inspiration from Pearl's framework.


- I'm not convinced that the causal framing adds value to this paper. The paper might be clearer if presented as a robust regression method. The model addresses an associational (level 1) prediction problem and does not involve interventions or counterfactual reasoning. While the terminology is "causal-inspired", the method itself does not perform causal inference. For instance, the derivation of the closed-form expression for $Y$ is probability algebra rather than causal calculus (e.g., backdoor adjustments or counterfactual reasoning).


- Clarify the true problem setting. If the experiments demonstrate robustness to corrupted labels (and not to distribution shifts, for example), then the rest of the paper should be written from this perspective. If not, you need additional experiments (e.g. feature-shifting experiments).


- Avoid philosophical overreaching. Replace slogans like "language of causality" or "interpretability-by-design" with precise technical descriptions. Avoid implying that the work establishes general causal robustness. Present your assumptions as pragmatic simplifications for analytic tractability and discuss their implications for generalization and where they might fail (e.g., nonlinear SCMs).

- Either remove interpretability claims or perform some qualitative analysis (e.g., visualizing $U$ representations and relating them to known latent factors).

- In the proposed future direction, you mention exploring "non-linear decision functions". Wouldn't this make $Y$ no longer Cauchy-distributed and thereby break the analytic tractability that the current framework relies on?

- Consider choosing a more descriptive name for the method. "Causal Regression" or "Causal Engine" sound very general, while the proposed method is better described by its linear mechanism and Cauchy-distributed uncertainty, e.g., "CauchyLinearRobustRegressor". Analogously, in logistic regression, the term "logistic" is part of the method name to convey what the method does and its underlying assumptions.



Minor:
- Types of noise (shuffle, asymmetric, outlier, systematic) are defined only in the appendix, though they are central to the paper.
- Reconsider how to introduce the Cauchy distribution in abstract. It's disconnected from the previous sentence.
- Show both MdAE and mean-based metrics.
- Figure 2 is too small to read. If it can't fit bigger in the main text, consider adding a LaTeX table in the main text and adding the figure in the abstract.
- Double dash: -— in line 100
- Double dash: —- twice in line 054-055
- Reword "we introduce two key assumptions", assumptions are made or adopted, rather than introduced.
- This question: "What distributional assumptions and functional forms are appropriate for causality?". "for causality" is too general, in practice, you're choosing assumptions that are appropriate for your particular setting, not for causality in general.
- From the paper:
"These results support our core hypothesis: by modeling the underlying causal graph, the Causal Engine is more effective at disregarding spurious correlations introduced by label noise than traditional models that rely solely on statistical associations.". In general, the method does not model the underlying causal graph. Also, the experiments don't demonstrate robustness to spurious correlations.

---

> ### Author Response · Authors · 2025-11-19
> **Clarifying Terminology: Pearl's Abduction vs. DiscoSCM Abduction**
>
> We have the utmost respect for your rigorous attention to causal terminology. Your critique regarding the "misuse of terms" is technically accurate within the context of classical SCM. However, as explained in our **General Response**, we are operating under the **DiscoSCM** framework, where these terms have distinct, defined meanings.
>
> **Response to "Misuse of causal terminology (Abduction)"**
>
> > *Critique: "The 'abduction' step is inference of a latent variable... not Pearlian abduction (inferring posterior over exogenous noise)."*
>
> You are correct that we are not performing Pearlian Abduction. We are performing **"DiscoSCM Individual Abduction"**.
> *   **Pearl's Abduction**: Infers $P(u_{noise} | e)$ to find the specific *random event*.
> *   **DiscoSCM Abduction**: Infers $P(U_{individual} | e)$ to find the *individual type* (e.g., patient identity).
> *   **Etymology**: In logic (Peirce), abduction is "inference to the best explanation." In our context, the "Individual Type $U$" is the best explanation for the observed features $X$. Thus, the term is logically appropriate, though distinct from Pearl's specific usage.
> *   **Action**: In the revised manuscript, we have added a dedicated **"Note on Terminology"** to explicitly clarify this distinction and avoid misleading readers familiar with Pearl's SCM.
>
> **Response to "Interpretability Claims"**
>
> > *Critique: "Intermediate variables Z, U, S are unlabeled... without semantic meaning, they remain uninterpretable."*
>
> We clarify that we claim **Structural Interpretability** (Modularity), not **Semantic Interpretability**.
> *   We do not claim to identify specific semantics for each dimension of $U$ (e.g., $U_1$=Age).
> *   We claim interpretability of the **inference process**: We can explicitly disentangle **Epistemic Uncertainty** (variance of $P(U|X)$, "I don't know this user") from **Aleatoric Uncertainty** (Cauchy scale, "This user's outcome is inherently noisy"). This ability to pinpoint the *source* of uncertainty is a form of interpretability that standard black-box models lack.
>
> **Response to "Philosophical Overreaching"**
>
> > *Critique: "Avoid philosophical overreaching... Replace slogans like 'language of causality'."*
>
> We appreciate your call for precision. However, we respectfully argue that the Cauchy distribution is not merely a "pragmatic simplification" but a **fundamental theoretical requirement** for our framework's Abduction step.
>
> *   **Why Cauchy?**: In our counterfactual world, any observation *could* theoretically be attributed to any individual type (the "everything is possible" principle). A light-tailed distribution (like Gaussian) mathematically forbids this by assigning effectively zero probability to extremes, violating the core philosophy of counterfactual inference.
> *   **Revision**: To address your concern about over-generalization while maintaining our theoretical stance, we will rename this hypothesis from "The Language of Causality" to **"The Language of Abduction"**. This more accurately reflects that the heavy-tailed Cauchy distribution is the specific mathematical language we use to describe the uncertainty in the **inference of latent individuals ($U$)**, rather than claiming it governs all causal phenomena.

---

> > ### Comment · Reviewer_Pd3p · 2025-11-24
> >
> > **Important**: The paper now exceeds the 10-page limit [1] allowed in the discussion period (currently 13 pages). I highly urge the authors to limit their submission to the allowed 10-page limit.
> >
> > [1] https://iclr.cc/Conferences/2026/AuthorGuide
> >
> > As part of my initial review, I wrote a point on "Misuse of causal terminology" contrasting Perlian terms with the ones used in the paper. The authors have now made it clear that they use DiscoSCM. Whether or not this resolves the issue depends on the next point on "causal framing". If causal inference is happening, natural questions arise on whether DiscoSCM is complete & sound, because it's not a peer-reviewed paper or a widely validated causal framework. Skipping for now, as I think this is not relevant, because the proposed method does not perform causal inference.
> >
> > > Causal framing
> >
> > I'm still not convinced that the causal framing adds value to this paper.
> > In its current form, the proposed method does not rely on causal inference machinery in the strict L2/L3 sense.
> > The model performs standard L1 prediction: given X, predict Y. The pipeline X->U->Y, the use of a latent variable U, additive noise $\epsilon$, linear maps, and heavy-tailed Cauchy likelihoods can all be formulated without appealing to DiscoSCM, abduction, etc.
> >
> > In other words, the architecture could be described entirely as a latent-variable robust regression model without invoking causal terminology, and its correctness would not change. From this perspective, the causal framing is interpretive rather than operational: it provides motivation, but it is not technically required.
> >
> > My concern is that the term "Causal Regression" may unintentionally lead readers to conclude that the method performs causal inference or answers interventional/counterfactual queries, which it does not. I still believe that the paper should clarify that the causal narrative is a conceptual motivation rather than a methodological requirement and that the model does not depend on the formal causal semantics of DiscoSCM or SCMs for its predictive performance.
> >
> >
> > > Interpretability
> >
> > The authors gave a clarification regarding "structural interpretability" vs "semantic interpretability." However, this distinction is vague and is not reflected in the current revision.
> >
> > The intermediate representations $Z$, $U$, and $S$ are unlabeled latent vectors with no demonstrated correspondence to meaningful factors in the data. These are not interpretable in the sense typically used in the ML literature (e.g., semantic grounding, visualization, or empirical evidence that they align with identifiable properties).
> >
> > The separation between epistemic and aleatoric uncertainty is useful, but it's uncertainty quantificationm not interpretability of the model's internal representations. The fact that the model has a modular structure does not by itself make the representations interpretable. It makes them structured, which is not the same thing.
> >
> > > Philosophical Overreaching
> >
> > My original comment was not about whether one should use Cauchy distributions, but about the rhetorical framing around causality. Cauchy priors indeed allow heavier tails and therefore (perhaps) more permissive inference than Gaussians. But **this is a modeling decision**.
> >
> > I cannot find any place in DiscoSCM that requires Cauchy specifically (or heavy-tailed priors), nor any theoretical argument in causal inference that makes Cauchy a "fundamental theoretical requirement" (copied from the authors' rebuttal). Any distribution with full support satisfies the principle that "anything is possible", so there is nothing uniquely causal about Cauchy in this respect. Using a Cauchy could perhaps be described as a well-motivated pragmatic decision, but not as a "fundamental theoretical requirement".
> >
> > Regarding the rebuttal saying it will rename the terminology: in the revised manuscript, the phrase "Language of Causality" still appears three times (L102, L292, L676, perhaps it's an editing error). However, I don't think that renaming "Causality" to "Abduction" addresses the central issue.
> >
> >
> > > Naming - CausalEngine
> >
> > The current name suggests that the method performs causal inference & it's too general. In practice, the model solves an L1 prediction task and does not carry out the full DiscoSCM pipeline (e.g., no valuation step, no counterfactual computation). A more descriptive name would better reflect the actual technical contribution and avoid implying causal capabilities that the method does not provide. As mentioned in the original rebuttal, a more descriptive name would be "CauchyLinearRobustRegressor". Analogously, in logistic regression, the term "logistic" is part of the method name to convey what the method does and its underlying assumptions.
> >
> > > Minor
> >
> > -$\textemdash{}$ on line 103
> >
> > > Final comment
> >
> > I still strongly believe my initial assessment of the paper is accurate.

---

> ### Author Response · Authors · 2025-11-26
> **On the Necessity of Causal Framing: No intervention $\neq$ no causation**
>
> We respectfully disagree with the reviewer's premise that causal framing is unnecessary without explicit intervention.
>
> **1. Structure Learning Is Causal, Not Just Statistical**
>
> The reviewer's implicit logic—"No intervention, no causation"—would, if taken to its conclusion, delegitimize the entire field of **Causal Discovery** pioneered by Pearl, Spirtes, and Glymour. Classic algorithms such as PC, FCI, and LiNGAM operate exclusively on observational data without any interventions, yet they are unambiguously part of causal inference. Our work shares this tradition: we aim to **recover** the latent causal structure ($U \to Y$) from data, which is an **identification** problem, not merely a prediction problem.
>
> **2. Abduction Is a First-Class Citizen of the Causal Hierarchy**
>
> In Pearl's three-layer causal hierarchy, **Abduction** is the foundational first step of **counterfactual reasoning (Layer 3)**, distinct from association-based prediction (Layer 1). Focusing solely on intervention (Layer 2) overlooks the fact that counterfactuals—and thus the robustness guarantees we claim—rest precisely on Abduction. We are not "avoiding" causality; we are operating at its most fundamental layer.
>
> **3. Generative Mechanism vs. Curve Fitting: The Newton Analogy**
>
> > *"Claiming our method is just 'Robust Regression' is like claiming Newton's Second Law ($F = ma$) is just 'Linear Regression' between Force and Acceleration."*
>
> Mathematically, both are linear relationships. But Newton's law assigns **causal semantics** to the variables: force *causes* acceleration. Similarly, DiscoSCM assigns the latent $U$ a precise causal interpretation—**Individual Type**—which is the invariant "root property" of each individual across environments. This structural commitment is what enables **OOD generalization**, not mere heavy-tailed robustness.
>
> Traditional robust regression focuses on statistical properties of $P(Y|X)$ (e.g., heavier tails via t-distributions). In contrast, we model the **Data Generating Process (DGP)** itself. The Cauchy distribution is not an "engineering hack for robustness"; it is derived from the DiscoSCM structural assumption about how individual heterogeneity enters the system (the "Everything is Possible" principle). Without the causal framework, the mathematical necessity of the Cauchy prior vanishes, and the model loses its explanatory power regarding *why* it generalizes.
>
> **4. On the "CauchyLinearRobustRegressor" Naming Suggestion**
>
> We appreciate the reviewer's concern about overly general naming. However, renaming to "CauchyLinearRobustRegressor" would be **misleading** because it:
>
> - Erases the causal semantics of $U$ that justify the model's invariance properties.
> - Reduces the contribution to a distributional choice, when the core novelty is the **structural causal model** itself.
> - Conflates our method with standard robust regression, which has fundamentally different assumptions and goals.
>
> We are open to alternative names that preserve the causal essence—e.g., **"DiscoSCM-L1"** or **"CausalIndividualRegressor"**—but we cannot accept a name that misrepresents the theoretical foundation.
>
> ---
>
> ###  On "The Language of Causality" Terminology
> We initially considered the revision to "The Language of Abduction" but have decided to retain the original phrasing. The reason is straightforward:
> > If the Gaussian distribution is "The Language of Statistics," then the Cauchy distribution is "The Language of Causality."
> This parallel is intentional and captures our core thesis—not philosophical overreach, but a principled correspondence.
> ---
>
> ### Summary
>
> | Reviewer Concern | Our Position |
> |------------------|--------------|
> | "No causal inference without intervention" | Causal Discovery (PC, FCI, LiNGAM) proves otherwise; Abduction is Layer 3 causality |
> | "Just robust regression" | We model the DGP mechanism, not just $P(Y\|X)$; $U$ has causal semantics |
> | "Cauchy is not theoretically necessary" | It follows from DiscoSCM's "Everything is Possible" assumption |
> | "Rename to CauchyLinearRobustRegressor" | This erases the causal semantics that justify the model |

---

> > ### Author Response · Authors · 2025-11-26
> > **Regarding the page limit**
> >
> > Thank you very much for your continued engagement and detailed feedback on our paper. We sincerely appreciate the time and effort you have invested in reviewing our work.
> >
> > We have submitted a revised version that now strictly adheres to the 10-page limit as required by ICLR 2026 guidelines. Thank you for bringing this to our attention.
> >
> > Regarding the causal framing: We would like to emphasize that our work presents the first computable end-to-end realization of DiscoSCM.

---

> > > ### Comment · Reviewer_Pd3p · 2025-11-26
> > >
> > > I strongly disagree with the authors.
> > >
> > > > Causal framing
> > >
> > > The authors' reply does not address the substance of my comment. The entire response is anchored on (and titled after) one sentence from my followup comment ("the proposed method does not rely on causal inference machinery in the strict L2/L3 sense"), which, while perhaps phrased imperfectly and is awkward without context, did not assert "no intervention = no causation", as the authors continuously emphasize. This is used to dismiss my underlying point, which remains valid. Below, I restate it more explicitly.
> > >
> > > *On causal discovery.*
> > > Causal discovery can act on L1 data, by making explicit causal assumptions and identifies structural information. In its simplest form: $X \to Y$ vs $Y \to X$ via conditional independencies (as some of the methods the authors mention).
> > > **This is not what CausalEngine does** (which is partly why the analogy to causal discovery in the rebuttal is misguided). There is no causal structure being *identified* in CausalEngine.
> > >
> > > *On causal representation learning.*
> > > Similarly, the model does not perform causal representation learning: $U$ has no factorization, modularity, or identifiability guarantees.
> > >
> > > *On the actual functionality of the method.*
> > > CausalEngine does not recover a graph, orient edges, or identify causal structure. **CausalEngine performs standard L1 prediction using a latent-variable architecture described in causal language**.
> > >
> > > The "L2/L3 machinery" comment concerned the use of DiscoSCM, which is a modelling framework that supports counterfactual reasoning, while the CausalEngine does not perform L3 reasoning or any form of causal identification. Removing the causal terminology from the paper would not change the method. The core issue, thus, is not the taxonomy of causal inference in my comment but that **the causal framing in the paper is interpretive rather than operational**. The model's correctness and robustness do not rely on DiscoSCM semantics (more on this in the Newton analogy), causal identification, or counterfactual reasoning.
> > >
> > > From the authors' reply: "We recover the latent causal structure $U \to X \to Y$ from data, which is an identification problem". This is incorrect. **The structure is not recovered, it's assumed**.
> > >
> > >
> > > > The Newton analogy
> > >
> > > I find the Newton analogy odd.
> > >
> > > *On the analogy.* $F=ma$ *is a linear function from acceleration to force*. Newton's law is not an algorithm/estimation procedure/regression model that infers the mass *m*, so it's not *linear regression* (as the authors state), but it is a linear function. It's not a statistical estimation procedure that recovers information from data. So its relevance here is questionable.
> > >
> > > I believe that the authors are trying to argue that $U$ inherits causal semantics from DiscoSCM. Are these semantics used (not as narrative motivation) in the proposed method? How are they useful? Note: if you just attribute empirical robustness to semantics, you'd need to specifically say how you are using those semantics.
> > >
> > > > Philosophical Overreaching
> > >
> > > The authors also attribute to me saying that Cauchy is an "engineering hack for robustness" which I have never said, and I do not appreciate the misquoting.
> > >
> > > The "Everything is Possible" principle motivates functions with full support, but full support does not uniquely identify the Cauchy distribution. Here is an incomplete list of functions with this property: cauchy, gaussian, laplace, logistic, student-t. Nothing forces the choice of Cauchy specifically or makes it **the "language of causality"** (quoted from the paper).
> > >
> > >
> > > > "We would like to emphasize that our work presents the first computable end-to-end realization of DiscoSCM"
> > >
> > > Does this work include the entire DiscoSCM pipeline (Abduction, Valuation, Prediction, Reduction)?

---

> ### Author Response · Authors · 2025-11-27
> **What We Actually Predict**
>
> We thank the reviewer for the continued engagement. We focus on three key points.
>
> ### 1. The DiscoSCM Pipeline and What We Actually Predict
>
> The reviewer asks: *"Does this work include the entire DiscoSCM pipeline (Abduction, Valuation, **Prediction**, Reduction)?"*
>
> First, the DiscoSCM pipeline consists of **three steps**, not four (Theorem 2, Lines 980-990). There is no separate "Prediction" stage.
>
> **Crucially, we must clarify what CausalEngine predicts.** We do **not** predict "what is this person's outcome $Y$?" (a point estimate). We predict:
>
> > **"For the subpopulation sharing observation $X = x_u$, what is the distribution of their counterfactual outcomes?"**
>
> This is fundamentally different from standard regression, and we believe this distinction may be the source of misunderstanding.
>
> **How CausalEngine's 4-stage architecture implements the 3-step theorem:**
>
> | DiscoSCM Step | CausalEngine Stage | Mathematical Operation |
> |---------------|-------------------|------------------------|
> | **Abduction** | Perception + Abduction | Infer posterior over Individual Types: $P(U\|X) = \text{Cauchy}(\mu_U, \gamma_U)$ |
> | **Valuation** | Action | Compute decision score with linear structural equations |
> | **Reduction** | Decision | Final **distributional** prediction |
>
> The 4-stage architecture is an engineering adaptation for the prediction task, but the mathematical essence is identical to the 3-step theorem. The output is a **Cauchy distribution** $(\mu_S, \gamma_S)$, not a scalar—this is why we use Negative Log-Likelihood loss, not MSE.
>
> ### 2. The Core Innovation: $U$ as Individual Type, Not Noise
>
> The reviewer asks: *"Are these [DiscoSCM] semantics used (not as narrative motivation) in the proposed method?"*
>
> **Yes, fundamentally.** The key innovation of DiscoSCM is the **explicit decoupling** of two sources of randomness that are **confounded** in classical SCM:
>
> | Framework | Latent Variable | Meaning |
> |-----------|-----------------|---------|
> | Pearl's SCM | $U$ (Exogenous) | Confounds individual identity with random noise |
> | DiscoSCM | $U$ (Individual Type) + $E$ (Noise) | **Decoupled**: $U$ = stable individual attributes; $E$ = environmental randomness |
>
> This changes **what Abduction means**:
> *   **In SCM**: Abduction asks "What random fluke caused this outcome?"
> *   **In DiscoSCM**: Abduction asks "**What kind of individual** am I looking at?"
>
> CausalEngine infers $P(U|X)$—the posterior over **Individual Types**—not over noise. This is why the model learns invariant representations and exhibits robustness.
>
> ### 3. Why Cauchy is Operationally Necessary
>
> The reviewer argues "nothing forces the choice of Cauchy." We contend Cauchy is the **unique solution** to two simultaneous constraints:
>
> **Constraint 1: Analytical Tractability (Stable Distributions)**
> For end-to-end differentiable inference without sampling, the distribution must be **Stable** (closed under linear combination). This restricts our choices. (Student-t is not stable.)
>
> **Constraint 2: No "Average Individual" (Undefined Mean & Variance)**
> Since $U$ represents **Individual Type**, we must capture the reality that **individuals cannot be represented by a population average**.
> *   **Gaussian (Defined Mean)**: Assumes a "representative individual" exists; others are deviations from it.
> *   **Cauchy (Undefined Mean & Variance)**: **Mathematically rejects the existence of an "average individual."** The sample mean never converges. This forces the model to treat every individual as a unique, irreducible causal entity.
>
> **Conclusion**: Cauchy is the **unique intersection** of {Stable} ∩ {Undefined Mean/Variance} as far as we known. If we used Gaussian, we would implicitly accept the "average man" fallacy. The Cauchy choice is the **operational signature** of the DiscoSCM axiom.
>
> ---
>
> *We also sincerely apologize for the earlier misattribution regarding "engineering hack."*

---

### Official Review · Reviewer_3K37 · 2025-11-05

**Soundness:** 2
**Presentation:** 2
**Contribution:** 2
**Rating:** 2
**Confidence:** 4

**Summary:**

The paper proposes a robust prediction framework called CausalEngine (CE), inspired by counterfactual inference (abduction, action and prediction). Given an input $X$, the model first extracts a representation $Z$ and infers a latent variable $U$ by modeling the posterior $p(U \mid Z)$ (similar to $P(U \mid X)$). The final prediction is then generated by applying a simple mechanism that maps $U$ (augmented with exogenous noise $\epsilon$) to the output $Y$. Both the latent cause $U$ and the noise term $\epsilon$ are modeled using Cauchy distributions. Experiments on both synthetic datasets and public regression benchmarks demonstrate that CausalEngine achieves improved robustness under various types of label noise.

**Strengths:**

**S1**. The paper includes many strong baselines, and the proposed CausalEngine (CE) consistently demonstrates robustness under various types of label noise.

**Weaknesses:**

**W1**. **Lack of theoretical justification**.

The paper lacks a theoretical analysis to support why CE should yield robust predictions. The current modeling and theoretical discussions are not sufficient to demonstrate that CE provides robustness guarantees. I strongly suggest formalizing the prediction problem and clearly stating under which assumptions CE is expected to produce robust results.

For example, consider the following SCMs:

$$X \leftarrow f_X(U_X), \quad Y \leftarrow f_Y(X, U_Y, \epsilon)$$

where $U_X$ is independent of $U_Y$. Then learning $P(U \mid X)$ in CE does not help robustness: $P(U \mid X)$contains no causal information about $Y$ beyond $X$ itself, and perturbing $\epsilon$ will still affect the prediction.

In contrast, if the SCM is

 $$X \leftarrow f_X(U_X, U_Y), \quad Y \leftarrow f_Y(U_Y, \epsilon)$$

then learning $p(U \mid X)$ may help, because $U_Y$ influences both $X$ and $Y$; inferring $U_Y$ could somewhat “recover” the direct cause of $Y$.

SCMs here are only illustrative; other formalizations are possible.
Grounding the method in clear assumptions would help readers understand when CE should be expected to improve robustness.

Additionally, since $U$ is unobserved, recovering the exact latent variable is generally impossible. **The paper should analyze whether CE can recover U up to an equivalence class**, i.e., whether different but equivalent representations still lead to robust predictions under the linear mechanism or Cauchy prior. Without such an analysis, the method may not generalize beyond the tested scenarios.

**W2**. **Limited experimental scope.**

The experiments involve only datasets with relatively low $X$ dimensionality (fewer than 100 features). It is unclear whether CE scales to high-dimensional inputs. Moreover, the paper assumes that the final mapping from $U$ to $Y$ is linear (line 93–98), but the public and synthetic datasets do not validate whether this assumption holds.

The introduction claims inspiration from linear reward heads used in LLM alignment:

*“Inspired by the success of reward modeling in large language models… where a simple linear head over a powerful representation can capture complexity…”*

Given this motivation, it would be compelling to test CE in a similar setting, e.g., using pretrained LLM embeddings as features for prediction on real-world tasks.

**Questions:**

**Q1**. Although the paper is motivated by causal concepts, the procedure seems to reduce to (1) learning a stochastic representation and (2) predicting from that representation. This appears similar to existing methods such as Variational Autoencoders (VAEs), which learn a latent stochastic representation $Z$ from $X$, and then perform downstream prediction tasks using $Z$. What is the key conceptual difference between CE and VAE-style latent representation learning?

---

> ### Author Response · Authors · 2025-11-19
> **Addressing Theoretical Justification & U-Identifiability**
>
> We appreciate your recognition of our method's "consistent robustness." We agree that without the **DiscoSCM** framework (detailed in our **General Response**), this robustness appears to lack theoretical backing. We hope the General Response has clarified the foundational theory. Here, we address your specific counter-examples and concerns.
>
> **Response to "Lack of theoretical justification" & Counter-Example**
>
> > *Your Example: If $X \leftarrow f_x(U_x), Y \leftarrow f_y(X, U_y, e)$ where $U_x \perp U_y$, then learning $P(U|X)$ does not help.*
>
> Your analysis is perfectly correct under standard SCM assumptions where $U$ represents independent noise. However, under **DiscoSCM**, the causal graph is fundamentally different:
>
> *   **The "Common Cause" Structure**: In DiscoSCM, $U$ represents the **Individual Type** (e.g., a patient's underlying health condition), which manifests as both symptoms ($X$) and outcomes ($Y$). Thus, the graph is typically **$X \leftarrow U \rightarrow Y$** (where $U$ is a confounder).
> *   **Necessity of Abduction**: In this structure, $U$ is the common cause. Inferring $P(U|X)$ is not just helpful; it is **necessary** to decouple the true heterogeneous causal signal from spurious noise in $X$. The value of inferring latent individual attributes ($U$) from high-dimensional history ($X$) is further supported by the recent published work in **Implicit Personalization** (e.g., *IP-Dialog*, EMNLP Findings 2025).
>
> **Response to "Recovering U" & Identifiability**
>
> > *Concern: "Recovering the exact latent variable is generally impossible... analyze whether CE can recover U up to an equivalence class."*
>
> We fully agree that recovering the "true" physical $U$ from observational data alone is theoretically impossible without further assumptions. However, strict identifiability is not required for robust prediction.
>
> *   **Inductive Bias as Causal Regularization**: Our core contribution is demonstrating that **when the learning process is guided by a framework with the correct inductive bias derived from causal theory (i.e., CausalEngine), the model can learn a representation $U$ that is more robust to spurious correlations in $X$.**
> *   **Architecture as Regularizer**: The CausalEngine architecture itself acts as a powerful regularizer. By constraining the mechanism to be invariant and heavy-tailed (Linear + Cauchy), it forces the model to discard spurious, complex patterns in $X$ and seek a representation closer to the causal essence.
> *   **Evidence**: We do not claim to perfectly recover the ground-truth $U$. Instead, our outstanding experimental results—where CausalEngine maintains performance while standard models fail under noise—are the most direct empirical evidence supporting this hypothesis: the model has successfully learned a predictively robust representation that mimics the behavior of the true causal variable.
>
> **Response to "Linearity Assumption"**
>
> > *Concern: "Assumes that the final mapping from U to Y is linear... severe limitation."*
>
> This design is intentional and does not limit the model's expressivity regarding $X$:
>
> *   **Global Non-Linearity**: The Perception module ($X \to U$) is a **deep non-linear neural network**. Therefore, the complete mapping from input $X$ to outcome $Y$ is highly non-linear and capable of modeling complex interactions in high-dimensional data.
> *   **The Linear Bottleneck as a Feature**: The linear constraint on $U \to Y$ acts as an **information bottleneck**. It forces the deep Perception network to perform all the complex non-linear extraction *before* the causal layer, condensing the messy $X$ into a compact, additive representation $U$. This ensures that $U$ captures high-level, stable factors rather than low-level noise.

---

### Official Review · Reviewer_3666 · 2025-11-05

**Soundness:** 2
**Presentation:** 3
**Contribution:** 3
**Rating:** 4
**Confidence:** 4

**Summary:**

The paper presents Causal Regression, a regression task that frames the target outcome as a function of latent variables that must be inferred from the given data. The solution presented is called CausalEngine, which uses a counterfactual reasoning process of perception, abduction, action, and decision to achieve this result. Notably the abduction step fits a posterior Cauchy distribution over the latent variables given the data, and the action step models the causal mechanism that maps the latent variables to the target. An extensive set of experiments is provided to test the robustness of the method in noisy settings.

**Strengths:**

1. The reframing of general regression tasks in causal terms is an interesting novel concept.

2. The assumptions of the Cauchy noise and the linear mechanism are clearly stated, justified, and emphasized as key components of the model.

3. The experiments are very comprehensive and demonstrate clear empirical advantages over baselines.

**Weaknesses:**

4. The causality aspect of the model seems undefined. While the action step is learning a “mechanism” for predicting the target, it is unclear how this is different from simply learning some kind of correlation between the latent variables and the outcome. There is no intervention being performed, it is unclear how confounding bias affects the system, and it is unclear how learning the latent variables mitigates this bias.

5. The claims that the abduction step learns latent variables and that the action step learns a mechanism is not supported by any identification results. It is unclear whether any of the learned results has any guarantees to be causally meaningful.

Overall, while I believe that the proposed method has interesting results, I do not quite understand what makes the model causal. If this is cleared up, I will raise my score.

**Questions:**

6. Does the distinction between epistemic and aleatoric uncertainty in Eq. 3 and Eq. 6 affect performance? Could the $\varepsilon$ term in Eq. 6 be omitted?

---

> ### Author Response · Authors · 2025-11-19
> **Clarifying the Causal Nature of CausalEngine & Answering Specific Questions**
>
> We sincerely thank you for your encouraging review. We are delighted that you recognized the comprehensive nature of our experiments and the clear empirical advantages of our method. Your primary question—"what makes the model causal"—is profound and central to our work. As outlined in our **General Response**, the answer lies in the **DiscoSCM** theoretical framework.
>
> **response to "What makes the model causal?"**
>
> You rightly observed that we do not perform explicit interventions (do-operators) during training. However, under the DiscoSCM framework, the "causal" nature of CausalEngine is defined by **what it learns (invariant mechanisms)** rather than **how it interacts (interventions)**.
>
> 1.  **Learning Invariant Mechanisms**: Standard regression often minimizes loss by exploiting spurious correlations ($P(Y|X)$). In contrast, CausalEngine is architecturally constrained to model the data generation process defined by DiscoSCM: $X \leftarrow U \rightarrow Y$. It forces the model to learn a representation of the **individual type $U$** (via Abduction) and the **stable mechanism** $f(U, \epsilon)$ (via Action) that generates $Y$.
> 2.  **Robustness as Evidence**: In causal theory, **invariance** to environmental shifts (like label noise) is the hallmark of a true causal mechanism. A model relying on spurious correlations would degrade significantly when those correlations are broken by noise. The fact that CausalEngine maintains high performance under severe noise (as you noted) is strong empirical evidence that it has successfully captured the underlying invariant causal structure ($U \to Y$).
>
>
>
> **Response to Specific Questions**
>
> > *Q1: Does the distinction between epistemic and aleatoric uncertainty in eq. 3 and eq. 6 affect performance? Could the ε term in eq. 6 be omitted?*
>
> This is a crucial design choice grounded in DiscoSCM theory, and our experimental comparisons of different CausalEngine modes (Endogenous, Exogenous, Standard) explicitly validate the contribution of both terms:
>
> *   **Endogenous Mode (Epistemic Uncertainty Only):** This mode models only the uncertainty of **"which individual type"** ($U$) we are observing, focusing on the variance in Eq. 3.
> *   **Exogenous Mode (Aleatoric Uncertainty Only):** This mode models only the **"intrinsic randomness"** ($\epsilon$) of the outcome via Eq. 6. The heavy-tailed Cauchy noise here is critical for absorbing large outliers.
> *   **Standard Mode (Both):** Our experiments demonstrate that the **Standard Mode**, which integrates both uncertainties, achieves the highest probability of superior performance across diverse noise settings. This suggests that Epistemic and Aleatoric uncertainties play distinct, complementary roles in handling noise: one hedges against the ambiguity of the individual's identity, while the other buffers against the inherent stochasticity of the outcome. Therefore, combining both terms provides a more robust defense against complex label corruption than relying on either in isolation.
>
>
> We hope this clarifies the causal grounding of our model. We have incorporated these explanations into the revised manuscript to prevent future confusion.

---

### Author Response · Authors · 2025-11-18
**Causal Regression: The First Computable End-to-end Realization of DiscoSCM**

**To the Esteemed Chair, Area Chair, and Reviewers,**

We are sincerely grateful for the insightful and remarkably consistent feedback from all reviewers. This feedback has led us to a crucial realization: our original manuscript suffered from a core narrative flaw. In our pursuit of conciseness, we failed to clearly articulate that our proposed CausalEngine is not an isolated algorithmic innovation, but rather the first computable and operational instantiation of a broader, more foundational theoretical framework we have been developing: **Distribution-consistency Structural Causal Models (DiscoSCM)**.

This **narrative gap** understandably led the reviewers to measure our work against the default yardstick of classical SCMs, resulting in profound and entirely valid questions regarding its causal claims, terminology, and design choices. We take full responsibility for this narrative shortcoming.

The central goal of this response is to bridge that gap. **First, we will clearly present the theoretical cornerstone of DiscoSCM. Second, we will demonstrate that every design choice within CausalEngine is a direct and principled operationalization of this theory.** We are confident that when our work is viewed within its proper theoretical context, all concerns about its causality will be resolved, its remarkable robustness will find a solid theoretical explanation, and its contributions will become significantly clearer.

To illustrate the shared nature of the reviewers' core concerns, we offer the following summary:

| Reviewer | Acknowledged Strengths | Shared Core Concerns |
| :--- | :--- | :--- |
| **3666** | "experiments are very comprehensive and demonstrate clear empirical advantages" | "do not quite understand what makes the model causal" |
| **3K37** | "consistently demonstrates robustness under various types of label noise" | "Lack of theoretical justification... why CE should yield robust predictions" |
| **Pd3p** | "robustness improvements for the chosen setting through comprehensive experiments" | "Misuse of causal terminology... not Pearlian abduction-action-prediction" |
| **Exq5** | "addresses a relevant and well-known limitation of standard machine learning models" | "unclear to me where the causal component lies within the proposed architecture" |
| **Consensus** | **The empirical results are convincing.** | **There are fundamental questions about the theoretical basis for the 'causal' claims.** |

---

> ### Author Response · Authors · 2025-11-18
> **Part 1**
>
> ### **Part 1: The Theoretical Cornerstone – DiscoSCM, a New Framework for Heterogeneity**
>
> Our research program begins with a fundamental question: How can we mathematically model **individual heterogeneity** in a way that is both elegant and powerful for reasoning? While traditional frameworks offer different abstractions, DiscoSCM provides a new answer by directly addressing the "degenerative counterfactual problem" inherent in models based on the strict consistency rule.
>
> The following table contrasts DiscoSCM with the Potential Outcome (PO) and Structural Causal Model (SCM) frameworks:
>
> | Feature | Potential Outcome (PO) | Structural Causal Model (SCM) | **Distribution-consistency Structural Causal Model (DiscoSCM)** |
> | :--- | :--- | :--- | :--- |
> | **Core Primitives** | Potential outcomes $Y(t)$ are treated as fundamental, indivisible entities. | Structural equations $vᵢ ← fᵢ(paᵢ, uᵢ)$ describe the data-generating physical mechanisms. | Individualized structural equations $vᵢ ← fᵢ(paᵢ, eᵢ; u)$ decouple individual attributes from the mechanism. |
> | **Fundamental Assumption/Theorem** | **Consistency Assumption**: $T=t ⇒ Y(t) = Y$. A core **assumption** linking observation to potential outcomes. | **Consistency Theorem**: $T=t ⇒ Y(t) = Y$. A **theorem** derived from the structural equations. | **Distribution-consistency Theorem**: $X=x, U=u ⇒ Y(x) \simeq Y$. A **theorem** derived from the DiscoSCM definition. |
> | **Individuality vs. Randomness** | Individuality is explicit in the unit index, relying on the SUTVA. Randomness arises from the sampling process. | Individuality and randomness are **confounded** in a single instance $u$ of the exogenous variable $U$. | Individuality and randomness are **explicitly decoupled**: $U$ for stable individual attributes (heterogeneity); $E$ for exogenous random noise (stochasticity). |
> | **Abduction Target** | **Subpopulation** defined by covariates. | The unobserved **exogenous noise** $u$. | The **individual representation** $U$. |
> | **Unit-Level Counterfactuals** | For an individual $u$, the potential outcome $Yᵤ(t)$ is a deterministic **fixed value**. | For an individual $u$, the counterfactual outcome $Yᵤ(t)$ is a deterministic **fixed value**. | For an individual $u$, the counterfactual outcome $Yᵤᵈ(t)$ is a **random variable** dependent on the counterfactual noise $E(t)$. |
> | **Handling of Noise** | Noise is not directly modeled but is part of the inter-unit variation. | The factual and counterfactual worlds share the **exact same** noise instance $u$. | The factual and counterfactual worlds' noises are **only identically distributed, not necessarily identical in value**, i.e., $E(x) \simeq E$. |
> | **Key Contribution** | Formalizes causal inference as a "missing data" problem. | Provides a language and graphical models for data-generating mechanisms, unifying the ladder of causation. | Solves the "degenerative counterfactual problem" at individual-level via "distribution-consistency," allowing for direct modeling of joint counterfactual distributions. |
> | **Framework Essence** | Views causal inference as a **missing data** problem. | Views causality as a **deterministic data-generating process** driven by exogenous variables. | Views causality as a **deterministic data-generating process** driven by exogenous variables **individualized (by $U$)**. |
>
>
> **The fundamental difference lies in two innovations:**
>
> 1.  **The Decoupling of "Individual U" and "Noise E"**: This is the core of DiscoSCM. In SCM, $u$ confounds the "identity" of an individual with "random chance." In DiscoSCM, $U$ represents stable, intrinsic attributes (the source of heterogeneity), while $E$ represents uncontrollable, circumstantial randomness.
>
> 2.  **A New Target for "Abduction"**: This decoupling imbues the classical "Abduction-Action-Prediction" inferential process with a new, more intuitive meaning, which is the very source of the reviewers' confusion.
>     *   In Pearl's SCM, "abduction" means inferring the state of the unobserved **exogenous noise** $u$.
>     *   In DiscoSCM, "abduction" means inferring the **individual type** $U$.
>
>     This is a profound distinction. SCM asks: "Given the symptom, what was the random fluke that started it?" DiscoSCM asks: "Given the symptom, what **kind of patient** am I looking at?" The latter is the bedrock of personalized reasoning.
>
> ---

---

> > ### Author Response · Authors · 2025-11-18
> > **Part 2**
> >
> > ### **Part 2: CausalEngine – An Algorithmic Realization of the DiscoSCM Theory**
> >
> > **CausalEngine is the first algorithmic implementation of this "Abduce (the individual type), then Predict" theoretical paradigm.** All of its design choices are precise and principled steps to make the abstract DiscoSCM theory operational.
> >
> > *   **What is the variable $U$ in our paper? (Response to all Reviewers)**
> >     *   It is **not** the exogenous noise of SCM. It is precisely the core primitive **$U$** from DiscoSCM theory, representing individual heterogeneity. The goal of CausalEngine is to learn an effective representation of this $U$.
> >
> > *   **What is the "Abduction" stage doing? (Response to Reviewer Pd3p)**
> >     *   It is **not** mimicking Pearl's abduction of "noise" in SCM. It is the **algorithmic implementation of the core DiscoSCM step of computing the posterior probability of the individual type, $P(u|e)$**. This perfectly explains why our "abduction" differs from SCM's—because it belongs to a different, more general theoretical system.
> >
> > *   **Why the "Linear-Causal + Cauchy-Noise" design? (Response to Reviewers Pd3p, 3K37)**
> >     *   This is not an arbitrary choice. It is a principled and practical design to make the DiscoSCM theory **computationally tractable**.
> >         *   **Principled**: The heavy tails of the Cauchy distribution perfectly match the philosophy of DiscoSCM's counterfactual world, where "anything is possible" (any observation has a non-zero probability of being attributed to any individual type).
> >         *   **Practical**: This combination yields "linear stability," allowing the entire inference chain from $P(U|X)$ to the final prediction to be computed **analytically**. This elegantly solves the primary computational hurdle in implementing DiscoSCM, avoiding costly and unstable sampling.
> >
> > *   **Where does the causality of CausalEngine lie? (Response to Reviewers 3666, Exq5)**
> >     *   Its causality is not found in performing a `do-operator`. It lies in its architecture **faithfully executing an inference process derived from a causal theory (DiscoSCM)**, a process fundamentally different from standard regression: $X \rightarrow P(U|X) \rightarrow Y$. The model learns not the brittle statistical association $E[Y|X]$, but the more stable causal mechanism $f(U, \varepsilon)$ described by DiscoSCM. **The exceptional robustness under label noise is the strong empirical evidence that it has successfully learned some form of this invariant mechanism.**
> >
> > *   **How do you guarantee that $U$ is "purely" causal? (Response to Reviewers 3K37, Exq5)**
> >     *   This is a profound and open question in causal representation learning. We do not claim to perfectly recover a "pure" ground-truth $U$ from observational data alone. Our core hypothesis and contribution is that: **when the learning process is guided by a framework with the correct inductive bias derived from causal theory (i.e., CausalEngine), the model can learn a representation $U$ that is more robust to spurious correlations in $X$**. In other words, the architecture itself is a powerful regularizer, forcing the model to seek a representation closer to the causal essence. Our outstanding experimental results are the most direct evidence supporting this hypothesis.
> >
> > ---

---

> ### Author Response · Authors · 2025-11-18
> **Part 3**
>
> ### **Part 3: Conclusion and Commitment to Revision**
>
> We acknowledge that our original manuscript failed to convey these crucial theoretical connections, which was the root cause of the reviewers' confusion. **To demonstrate our commitment to addressing these issues, we have uploaded a revised manuscript as part of this rebuttal.** In this revision, we perform a series of "surgical" modifications to reshape the narrative, guided by the principle of minimal necessary change:
>
> 1.  **Plant a seed in the Abstract**: We will revise a sentence in the abstract to explicitly state that CausalEngine is an operationalization of the DiscoSCM theory, immediately setting the correct context. For example: "We introduce CausalEngine, a neural architecture that operationalizes this paradigm based on the Distribution-consistency Structural Causal Model (DiscoSCM)."
>
> 2.  **Establish a "Signpost" in the Introduction**: While preserving the original narrative flow that begins with a concrete regression problem, we will insert a crucial paragraph. This "signpost" will briefly introduce the limitations of classical SCMs in handling individual heterogeneity and point to DiscoSCM as the theoretical solution our work builds upon.
>
> 3.  **Reframe the "Related Work" Section**: Instead of creating a new "Preliminaries" section, we will embed the detailed comparison table of the three causal frameworks (PO, SCM, DiscoSCM) directly within the "Related Work" section. This will allow us to situate our work and justify our choice of DiscoSCM in a single, cohesive argument.
>
> 4.  **Correct the Mathematical Provenance**: In the Methodology section, we will replace the reference to Pearl's abduction-action-prediction process with the specific "Population-Level Valuation" theorem from DiscoSCM. This will clarify that our CausalEngine's three-step inference is a direct algorithmic implementation of this theorem's mathematical derivation (Abduction $\to$ Valuation $\to$ Reduction), rather than a loose adaptation of the SCM framework.
>
> By implementing these targeted, surgical revisions, we have effectively aligned the manuscript with its true theoretical foundation without disrupting the narrative flow that the reviewers found compelling.
>
> We firmly believe that this clarification illuminates the core contribution of our paper: **we have not only proposed a robust regression algorithm that excels in practice, but more importantly, we have successfully built a bridge between a deep causal theory (DiscoSCM) and practical machine learning, paving the way for this theory's future development and application.**
>
> We thank the reviewers again for their invaluable feedback, which has helped us to fundamentally improve the value and clarity of our work.

---

### Note · Program_Chairs · 2026-01-17
**Submission Desk Rejected by Program Chairs**

The following references in this submission do not refer to real documents and/or have major errors in bibliographic information:

 Jie Zhong, Weihua Shen, Yaliang Li, and Dacheng Tao. A comprehensive survey of instructionfollowing language models: Architectures, applications, and future trends. arXiv preprint arXiv:2504.12328, 2025